# Transcriptome analysis unravels the biocontrol mechanism of *Serratia plymuthica* A30 against potato soft rot caused by *Dickeya solani*

**Iman Hadizadeh**[1]*, **Bahram Peivastegan**[1], **Kåre Lehmann Nielsen**[2], **Petri Auvinen**[3], **Nina Sipari**[4], **Minna Pirhonen**[1]

1 Department of Agricultural Sciences, University of Helsinki, Helsinki, Finland, 2 Department of Chemistry and Bioscience, Aalborg University, Aalborg, Denmark, 3 Institute of Biotechnology, University of Helsinki, Helsinki, Finland, 4 Faculty of Biological and Environmental Sciences, Viikki Metabolomics Unit, University of Helsinki, Helsinki, Finland

* ihadizadeh@gmail.com

**Data Availability Statement:** Following the guidelines provided on the PLOS ONE website, both RNA-Seq datasets were deposited into the

## Abstract

Endophytic bacterium *Serratia plymuthica* A30 was identified as a superior biocontrol agent due to its effective colonization of potato tuber, tolerance to cold conditions, and strong inhibitory action against various soft rot pathogens, including *Dickeya solani*. We characterized transcriptome changes in potato tubers inoculated with *S. plymuthica* A30, *D. solani*, or both at the early and late phases of interaction. At the early phase and in the absence of the pathogen, A30 influenced the microbial recognition system to initiate plant priming. In the presence of the pathogen alongside biocontrol strain, defense signaling was highly stimulated, characterized by the induction of genes involved in the detoxification system, reinforcement of cell wall structure, and production of antimicrobial metabolites, highlighting A30's role in enhancing the host resistance against pathogen attack. This A30-induced resistance relied on the early activation of jasmonic acid signaling and its production in tubers, while defense signaling mediated by salicylic acid was suppressed. In the late phase, A30 actively interferes with plant immunity by inhibiting stress- and defense-related genes expression. Simultaneously, the genes involved in cell wall remodeling and indole-3-acetic acid signaling were activated, thereby enhancing cell wall remodeling to establish symbiotic relationship with the host. The endophytic colonization of A30 coincided with the induction of genes involved in the biosynthesis and signaling of ethylene and abscisic acid, while downregulating those related to gibberellic acid and cytokinin. This combination suggested fitness benefits for potato tubers by preserving dormancy, and delaying sprouting, which affects durability of tubers during storage. This study contributes valuable insights into the tripartite interaction among *S. plymuthica* A30, *D. solani*, and potato tubers, facilitating the development of biocontrol system for soft rot pathogens under storage conditions.

Sequence Read Archive (SRA) with the accession numbers PRJNA1085008 and PRJNA1085537 for the time course and second RNA-Seq datasets, respectively.

**Funding:** This work received APC funding from the University of Helsinki [FI0313471].

## Introduction

Potato (*Solanum tuberosum*) is one of the most promising nongrain crops in the world for food security and sustainable agriculture because of its superior nutritive qualities, high yields in a short time, and diverse distribution pattern worldwide [1]. However, various environmental stresses and pathogenic organisms inflict significant losses to vegetating and ware potatoes [2]. Among these, the phytopathogenic bacterium *Dickeya solani*, as a highly virulent member of soft rot *Pectobacteriaceae* (SRP), is responsible for blackleg and tuber soft rot of potato in many potato-growing regions with severe economic loss in both field and storage [3]. In Europe, the main losses are linked to the downgrading of potato seed lots and reducing of the market value [4]. SRP bacteria attack their host cells by the production of cell wall-degrading enzymes causing tissue maceration, and hence, they have long been considered to be brute force necrotrophic pathogens. However, SRPs like most gram-negative bacteria can be described as hemibiotrophic pathogens, exhibiting two distinct infection phases. The initial phase is biotrophic, allowing bacteria to latently proliferate within host tissue during the asymptomatic stage prior to switching to the symptomatic (necrotrophic) phase. The onset of latent infection is largely depending on the environmental conditions and host physiological status that affects bacterial growth, and in this way, bacteria are transmitted to the next generations through seed tubers [3,5].

Effective management of soft rot in stored tubers is challenging due to the lack of efficient measures against these pathogens. Since the disease is primarily seed-borne, control relies on a protocol for detecting the bacteria during certification of seed potatoes, applying hygienic measures, and utilizing physical treatments to control tubers contamination [2]. Nevertheless, only partial control of the disease has been achieved because these are limited possibilities for a consistent reduction of the bacterial inoculum level on the seed tubers has limited possibilities. The development of post-harvest biological control methods for soft rot offers promising environmentally friendly strategies to improve potato health, and prevent disease across various cultivation or storage conditions [6]. Potato roots and tuber surfaces host a diverse and dense microbial community that constitutes a valuable source for selecting biocontrol agents in the potato ecosystem [7]. Several studies have described encouraging results on the efficiency of biocontrol strains against SRP species in artificial growth media and plant tissues under laboratory conditions [8–10], although limited experiments have been performed under greenhouse, field or storage conditions for the evaluation of biocontrol agents directed against *D. solani* [8,11,12]. The endophytic bacterium *Serratia plymuthica* strain A30 has demonstrated unique capabilities in preventing soft rot disease both in vitro and on potato plants. This strain produces antibiotics, biosurfactant, and plant growth-promoting auxins [8,13]. The *S. plymuthica* A30 is able to stably colonize potato roots and stems in the xylem vessels and between parenchyma cells. During co-inoculation with *D. solani*, A30 was observed inside both roots and stems after 7 days, with no presence of pathogen cells, indicating considerable activity toward *D. solani*. When applied to seed tubers, its rapid colonization of surface and internal tuber tissues is attributed to competition for niches and space [13,14]. In our previous research, A30 was identified as the best low-temperature tolerant strain capable of surviving and colonizing potato tubers at a high population density during long-term storage and the following season in the field. Pre-treating tubers with this antagonist led to a rapid decrease in the population density of *D. solani*, indicating the effectiveness of strain A30 in reducing soft rot occurrence during storage and transfer to tuber progeny, as well as the incidence of blackleg during field cultivation of daughter tubers [12].

Biocontrol strategies rely on antagonists that directly affect pathogen populations via antibiosis, competition for nutrients and space, and production of secondary metabolites

(antioxidant and hydrolytic enzymes, toxins, antibiotics, siderophores, and phytohormone), while the interference with quorum sensing, and/or the induction of host resistance are indirect mechanisms against pathogens [15]. Knowledge about mechanisms of action for most biological control agents is relatively superficial, primarily due to a lack of understanding of the key biochemical and molecular processes occurring within multidimensional interactions of the host to microorganisms (beneficial and pathogenic). Dissecting relationship of this nature goes beyond isolation and characterization of a single or small number of genes or proteins and in where "-omics" supported research can make large strides. Only a few experiments have been executed to understand the response mechanisms of potatoes challenged with beneficial microbes using transcriptomic techniques [16,17].

Beneficial microbes can prime plant defense responses against various pathogens by activation of induced systemic resistance (ISR). This resistance contrasts with the systemic acquired resistance (SAR) correlating with the involvement of salicylic acid (SA) and the activation of pathogenesis-related (PR) genes. Several endophytic bacteria, mainly in genera *Bacillus* and *Pseudomonas*, have been shown to protect host plants using defense priming by ISR [18]. Extensive studies have demonstrated that ISR operates through multiple phytohormone pathways to regulate the defense signaling network. Plant hormones SA, Jasmonic acid (JA), and ethylene (ET) may be involved in the endophytic priming of defense, however, the predominance of each signaling pathway strongly depends on the players that are interacting [15,18]. Endophytic *Actinobacterium* exhibited the ability to prime different induced resistance against two different pathogens, as the resistance to *Erwinia carotovora* mediated through the JA/ET pathway, while resistance towards *Fusarium oxysporum* involved the SA pathway [19]. In grapevine, both *Bacillus subtilis* and *Pseudomonas fluorescens*-induced ISR involve the activation of JA/ET-dependent defenses, as prime immune pathways against *Botrytis cinerea* [20]. Conversely, *B. subtilis*-induced ISR against *Pseudomonas syringae pv. tomato* DC3000 relies on the SA and JA/ET pathways, while ISR induced by *P. fluorescens* is independent of the SA pathway [21]. Activated ISR by *Bacillus cereus* AR156 triggered both SA and JA/ET signaling pathways against *Pst DC3000* in Arabidopsis, while requiring only the JA/ET pathways against *Botrytis cinerea* [22]. Nevertheless, the dependence of ISR on JA/ET pathways is widely supported by numerous studies [23,24].

Microbes, including endophytes and pathogens, can be detected by host pattern recognition receptors (PRRs) that respond to microbe- or pathogen-associated molecular patterns (M/PAMPs) or host-derived molecular signals consisting of damage-associated molecular patterns (DAMPs), which activate the immune defense signaling response termed pattern-triggered immunity (M/PTI). When endophytes and pathogens invade resistant plants, the virulence effectors are translocated to overcome the first layer of plant immunity and, in turn, stimulate an amplified and robust form of defense termed effector-triggered immunity (ETI). The successful activation of M/PTI and ETI induces plant defense against invading microorganisms by changing the ion flux across the plasma membrane, increasing cytosolic $Ca^{2+}$ levels, activating mitogen-activated protein kinase (MAPK), and accumulating apoplast reactive oxygen species (ROS), followed by manipulation of phytohormones signaling, cell wall remodeling and defense-related transcriptional and metabolic reprogramming [25]. The gatekeeping functions of the plant immune system are capable of distinguishing pathogens from beneficial microbe through screening MAMPs. Establishing symbiosis requires a mechanism to avoid or circumvent plant immune responses. Beneficial endophytes employ various strategies to either evade PRR recognition or interfere with immunity signaling. The latter is accomplished by secreting weak effectors that minimize stimulation of the host's immune system and allow them to successfully associate with their host plants [26]. The dynamic relationship between endophytes and hosts, maintaining a delicate balance between endophytic virulence factors

and the plant defense system, is explained by the "balanced antagonism" theory. As the host colonization progresses and nutrient exchange initiates, the plant gradually perceives the microbe as a friend [27]. Characterizing the transcriptional crosstalk triggered by the pathogen and the beneficial microbe in host tissue could lead to the design of more effective biocontrol strategies.

The objective of this research was to elucidate the transcriptional responses of potato tubers interacting with the beneficial bacterium *S. plymuthica* A30 and the pathogenic bacterium *D. solani* during the early and the late phases of interaction, to describe how beneficial microbes influence plant fitness and defense signaling pathways. Our results revealed that A30 triggers defenses signaling including recognition and cell wall strengthening, as well as induced-resistance through JA signaling during the early phase of interaction. In the late response, A30 actively suppressed plant immunity, while promoting cell wall remodeling and indole-3-acetic acid (IAA) signaling that facilitate colonization phase. It prolonged dormancy in tubers by inducing dormancy-related hormones and suppressing growth-related hormones.

## Materials and methods

### Plant material and bacterial inoculation

Potato tubers of the cultivar 'Bintje', susceptible to blackleg (*Pectobacterium* spp.) (http://www.europotato. org/menu.php), were used in RNA-Seq profiling experiments and obtained from Finnish Seed Potato Centre Ltd. Potato cultivar 'Lady Felicia' was used throughout other experiments and purchased from a wholesale company (H&H Tuominen Ltd.). All the tubers were tested negative for *Dickeya* and *Pectobacterium*. A highly aggressive Finnish *D. solani* isolate (strain Ds0432-1) and its biocontrol agent *S. plymuthica* A30 were obtained from the bacterial collection of the Plant Pathology Laboratory of the University of Helsinki, Finland. Both bacterial strains were grown in Luria-Bertani broth at 28°C with agitation (220 rpm) for 24 hours, and bacterial cells were collected and re-suspended in sterile double-distilled water with the concentration adjusted to 0.2 in OD600 corresponding to about $5 \times 10^6$ colony forming units per milliliter (CFU ml$^{-1}$) for *D. solani* and about $1 \times 10^9$ CFU ml$^{-1}$ for *S. plymuthica* A30, as determined by plate counting.

### Tuber inoculation, sample preparation, and RNA isolation

Potato tubers were washed in tap water, surface sterilized by immersion in 3% sodium hypochlorite for 5 min and dried for 45 min at room temperature in the dark before inoculation. Each tuber was wounded at two sites spaced uniformly (~3 cm apart) across the tuber surface, with a sterile cork borer creating two wounds approximately 5 mm in diameter and 10 mm deep. First, a group of tubers was inoculated with 25 μl suspension of strain *S. plymuthica* A30 and another group with 25 μl of sterile water as non-inoculated control. After two hours, non-inoculated controls and tubers treated with strain A30 were inoculated with 25 μl suspension of *D. solani* or an extra 25 μl of water, depending on the treatments. Thus, wounded tubers inoculated with 50 μl of sterile water were maintained as control. Each hole was covered by Vaseline to avoid the healing process and the tubers were placed on top of stainless sieves in a plastic box internally lined with moist cloth to keep high humidity (80–90%). The boxes were then sealed with a plastic lid and incubated at 15°C in the dark. For RNA-Seq analysis, inoculated tubers were cut in half and tissue (0.5 cm × 0.5 cm) was cut out from healthy tissue beside the inoculation point by a scalpel. The sample included cells from the periderm, cortex and medulla layers. The samples were frozen immediately in liquid nitrogen and stored at -80°C. The total RNA of each sample was extracted using the CTAB method [28]. TURBO DNA-free kit (Ambion) and RNeasy MinElute clean-up kit (Qiagen) were used for DNase treatment and

RNA purification, respectively, according to the manufacturer's instructions. RNA purity and quantity were assessed using a NanoDrop ND-2000 spectrophotometer (Thermo Scientific, Wilmington, DE, USA) and RNA integrity was further checked on a 2100 Bioanalyzer Instrument (Agilent Technologies, Santa Clara). Ribo-zero kit was used to remove unwanted ribosomal RNA (rRNA) and libraries were constructed by high-throughput TruSeq RNA Sample Preparation protocol according to the manufacturer's instruction. Five groups of four tubers were randomly selected for each treatment at different time points, and samples were collected from three biological repetitions and three technical replicates. The experiment was repeated three times with new batches of tubers and inoculums. The same sampling method was used for phytohormone measurement, each in a separate experiment.

## RNA-Seq experimental design and data analysis

A graphical representation of the experimental design is presented in Fig 1. To design a high-resolution expression experiment, two RNA-Seq experiments were carried out. In total, the following combinations were analyzed: potato tubers treated by *S. plymuthica* A30 only (+A30−D), tubers inoculated by *D. solani* only (−A30+D) and tubers treated with both bacteria (+A30+D), all compared to control tubers treated with water (−A30−D). In the first RNA-Seq, a time course experiment was conducted using the Illumina HiSeq 2000 platform. Eighteen samples were examined, comprising three time points [1-, 24-, and 168-hours post-inoculation (hpi) with *D. solani* or an additional water (second inoculation)], three biological replicates, and two treatments (+A30−D and +A30+D), in comparison against water-inoculated tubers (−A30−D). This study provided an overview of transcriptional changes in the host response to bacterial biocontrol over the course of the interaction. In the second RNA-Seq experiment, the NextSeq 500 platform was used to analyze potato tuber transcriptional responses at 24 hpi to better understand the mechanisms underlying the resistance induced by strain A30 against *D. solani*. The transcriptional profiles were compared, considering the pathogen as a factor, between the groups of A30-inoculated (+A30+D) and non-inoculated (−A30+D) tubers.

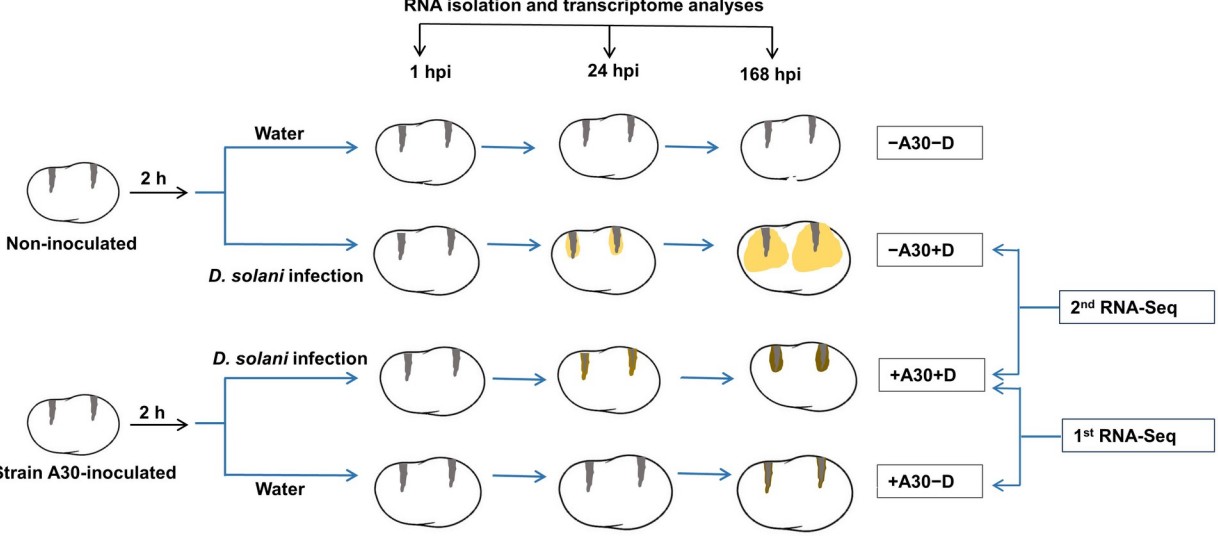

**Fig 1. Graphical representation of the experimental design.** Tubers were either inoculated with *S. plymuthica A30* (+A30−D) or left non-inoculated (-A30−D). Two hours later, a subset of tubers, both from A30-inoculated and non-inoculated groups, were exposed to soft rot pathogen *D. solani*, designated as (+A30+D) and (−A30+D) respectively. Subsequently, sampling and analysis were conducted at 1-, 24-, and 168-hours post-inoculation.

Through base calling, the original image data from the sequencing platform were transferred into the corresponding nucleotide sequence data FASTQ file. FASTQC (V0.10.1) was used to determine the quality of the RNA-Seq data (http://www.bioinformatics.babraham.ac.uk/projects/fastqc/). Prior to mapping reads to the reference database, the raw sequencing reads underwent a rigorous pre-processing step including eliminating those with low-quality sequences, containing more than 5% unknown nucleotides, and lacking Illumina adaptors. At the same time, quality scores (Q20, Q30), GC content and sequence duplication level were calculated for the cleaned data. The Q20 and Q30 values corresponded to base call accuracies of 99% and 99.9%, respectively. All subsequent analyses relied on this refined set of high-quality data. The Potato Genome Sequencing Consortium (PGSC) sequence of the doubled monoploid *S. tuberosum* Group Phureja DM1–3516 R44 with Genome Annotation v3.4 (PGSC_DM_v4.03) was used as reference data (http://potato.plantbiology.msu.edu/pgsc_download.shtml) as described earlier [29].

In summary, the potato genome sequence (PGSC_DM_v4.03) served to build the mapping index using Bowtie2 (v2.1.0). Subsequently, clean reads were aligned to this indexed genome using TopHat (v2.1.1). The mapping parameters were "-p 8 —b2-very-sensitive—solexa-quals —segment-length 30—segment-mismatches 3—mate-std-dev 20—library-type fr-unstranded" (http://ccb.jhu.edu/software/tophat/index. shtml). Cufflinks (v2.2.1) was employed with default settings [30], to compute the relative abundances and variations between treatments. The expression level of each transcript was determined based on the fragments per transcript kilobase per million fragments mapped value (FPKM). The correlation coefficient of $> 0.90$ was performed in each library and the cut-off value was the 95% confidence limit of the FPKM for all genes. Statistically significant differentially expressed genes (DEGs) were identified by NOISeq (V2.28.0) analysis with the R/Bioc package using read counts with probability $> 0.75$, p-value $< 0.05$ and log2 fold change $\geq 9$ and $\leq -9$. A visual inspection of the samples using principal component analysis (PCA) was followed by combining the technical replicates of each sample by averaging their gene counts for subsequent analysis steps. Furthermore, Venn diagram analyses were drawn by BioVenn software to determine unique and overlapped DEGs in the treatments. To visualize gene expression, hierarchical cluster analysis was carried out by Cluster 3.0 and Java TreeView software (v1.1.6r4). All identified transcripts were subjected to enrichment analysis to explore their associations with the Gene Ontology (GO) and Kyoto Encyclopedia of Genes and Genomes (KEGG) database, aiming to unveil their biological functions and involvement in functional pathways. Subsequently, GO annotation and functional classifications were assigned utilizing Blast2GO and WEGO software. Metabolic pathway annotation and enrichment of the DEGs were performed by using KEGG (KEGG- http://www.genome.jp/kegg/). GO and KEGG terms with a corrected p-value $< 0.05$ were considered significantly enriched for the DEGs. To carry out detailed gene sorting for interpretation of RNA-Seq data in terms of general physiology, NCBI RefSeq annotations of potato genes were used to our RNA-Seq data. The transcripts were subjected to BLASTX searching against the NCBI non-redundant (Nr) database with a typical cutoff E-value $\leq 10^{-5}$. The classification based on the above-mentioned databases was created, manually checked, edited, and enriched by numerous sources of literature. Gene categories were manually subdivided into different subcategories.

In the second RNA-Seq, the clean data with high quality were mapped to genome and gene references of the potato reference genome (PGSC_DM_v4.03) using the Burrows-Wheeler Alignment Tool (BWA v0.6.2) and Bowtie, respectively. Following alignments, the gene expression level was quantified by RSEM (RNA-Seq by expectation maximization) software tool, and the most highly significant DEGs were found by the criterion of the absolute value of RPKM ratio $> 1,000$. The EBSeq, NOISeq (V2.28.0) package methods, and Poisson distribution analysis method were carried out to retrieve the DEGs list. NOISeq-sim was used with the highest threshold ($q \geq 0.8$) to compute the probability of gene expression in comparisons. The

normalization techniques RPKM, Upper Quartile, and Trimmed Mean of M values (TMM) were implemented in NOISeq. Blast searches were conducted against NCBI and Arabidopsis Information Resource (TAIR10) to seek the homologs with *S. tuberosum* and *Arabidopsis thaliana*. DEGs were identified in 24 hpi for two comparison groups: (+A30+D) vs. (−A30−D) and (−A30+D) vs. (−A30−D). For functional enrichment analysis of the DEGs, GO terms were assigned to the assembled transcripts and the GO annotation results were explored by Blast2GO. Likewise, the KEGG database was used to visualize the DEG dataset and to further illustrate the gene functions. The significantly enriched function terms were identified using the criterion of a Bonferroni-corrected $p \leq 0.05$.

## Validation of DEGs via real-time qPCR

The Real-time qRT-PCR analysis was performed to validate transcriptome profiling results achieved by the two RNA-Seq experiments. The primers used are listed in S4 Table. Primer-Quest tools (http://www.eu.idtdna.com/Primerquest/ Home/Index) from Integrated DNA Technologies were used to design the primers. The sequence transcripts were retrieved from the transcript reference file (PGSC_DM_v4.3_transcript_representative.fasta.zip) from the SOL genomic network. To make cDNA, clean total RNA (1 μg) was reverse transcribed by Enhanced Avian Reverse Transcriptase (Sigma, A 4464) according to the manufacturer's instructions. An epMotion® 5075 pipetting robot (Eppendorf) was used for all PCR reactions pipetting and the qPCR experiments were performed on a LightCycler® 480 Real-Time PCR System (Roche) with three replicates of independent cDNAs. The ubiquity of the qPCR amplicon of each target gene was checked on the blastx and then visualized on 1% agarose gel to verify the amplification of a single product with the expected length. The expression value of the target genes was calculated by $2^{-\Delta\Delta CT}$ based on the Livak method [31], and "eukaryotic initiation factor 5A" was used as a housekeeping gene [32] to normalize the relative fold changes of expression values. The qRT-PCR was carried out in three independent experiments with three technical replicates of each cDNA, with similar results, and the data for each experiment were analysed separately.

## Extraction and quantification of plant hormone metabolites

To investigate the responses of potato tubers to strain A30 and to evaluate the impact of the interaction on hormone homeostasis, the levels of SA, inactive SA 2-O-beta-D-glucoside (SAG) (stored form of SA), JA, JA precursor 12-oxo-phytodienoic acid (OPDA), abscisic acid (ABA), and IAA were determined under different treatments. The contents of the hormone metabolites were analyzed by Ultra performance liquid chromatography-tandem mass spectrometer (UPLC-MS) system. Approximately 400 mg fresh weight (FW) of samples in four treatment groups +A30-D, +A30+D, -A30+D, -A30-D, and two time points 24 and 168 hpi were extracted as described in Peivastegan et al. [29]. Five replicate samples, each containing ground tissue from five treated tubers, were analysed in the experiments. The results of SA, JA, OPDA, ABA and IAA were normalized to the corresponding deuterated ISTDs (SA-d4, dh-JA, DnOPDa-d5, ABA-d6 and IAA-d5, respectively) and FWs were quantified using calibration curves for each phytohormone with Analyst MultiQuant™ software (ABSciex Pte. Ltd.).

## Results

### Profiling potato tubers transcriptome changes in response to *S. plymuthica* A30 with or without *D. solani*

A time-course transcriptome analysis was conducted to identify the specific host response to strain A30 both before and after infection with *D. solani* at three time points: 1 and 24 hpi,

which correspond to the initial bacterial growth and host recognition, and 168 hpi, which aligns with endophytic bacterial development and colonization. To determine the relationship between the different treatments and their three biological replicates, the samples were clustered by PCA. The control and treatment groups were separated with the most variance on the x-axis (principal component 1, PC1) (44.6%). The samples from different time points could be separated by the second axis as the y-axis in which 168 hpi separated from earlier time points with less variance in the data (principal component 2, PC2) (25.2%) (Fig 2). Separation was also observed in each treatment group in a time-dependent manner. The largest separation was observed at 24 hpi, whereas the samples were more similar to each other at 186 hpi.

The number and function of DEGs for each comparison at the early and late time points are detailed in S1 Table. In the absence of the pathogen, strain A30 had a subtle effect on the host gene expression profile, causing expression change in about 6.1% of the potato transcriptome. In contrast, treatment with both A30 and *D. solani* induced rapid and more pronounced transcriptional changes (Fig 2). Application of *S. plymuthica* A30 triggered a high number of upregulated genes detected 24 hours after inoculation with *D. solani*, highlighting the complex mechanism of A30-induced resistance. RNA-Seq analyses of treated tuber samples collected at the late time point (168 hpi) revealed an increase in the total number of DEGs with

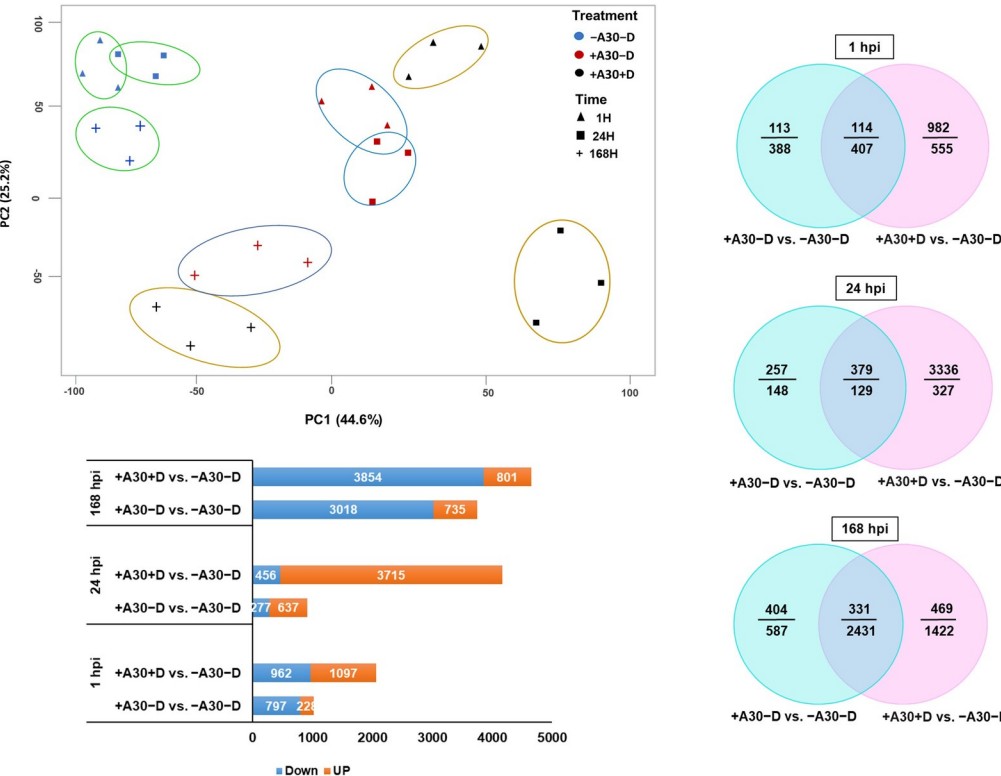

**Fig 2. Transcriptomic changes in potato tubers treated by *S. plymuthica* A30 before and after challenge with *D. solani* at different time points.** Principal Component Analysis (PCA) was conducted on gene expression data obtained from potato tubers subjected to different treatments: Non-inoculated control tubers (-A30-D), tubers inoculated with a single strain A30 (+A30-D), or tubers inoculated with two strains (+A30+D). Treatments are represented by colored dots, while sampling times are denoted by different shapes. Each color represents a treatment group, and each shape within a color represents one of the three biological replicates for each treatment. Number of differentially expressed genes (DEGs) that were either up- or down-regulated in each comparison at different time point (adjusted p < 0.05). Venn diagrams present pair-wise overlap of up- and down-regulated DEGs in each treatment group compared with mock-inoculated tubers at different time points.

considerable overlap in both the numbers and identity of genes between treatments by single strain A30 and combined strains (pathogenic and biocontrol). However, the response was notably milder in the presence of only A30 compared to the combined treatment. The Venn diagram represents shared DEGs between the two treatments throughout the time course (Fig 2). Although the total number of DEGs was relatively lower in the +A30−D compared to +A30 +D, a substantial portion of responsive genes in +A30−D showed the same expression pattern as in +A30+D. This similarity could be attributed to the presence of strain A30. To gain an insight into the function of the expressed genes, GO enrichment analysis of highly expressed genes was applied (S1 Table). Our primary focus centered on GO terms within the biological process (BP), categorized into "Response to stimulus," "Cell communication," "Cellular process," "Metabolic process," "Developmental process," "Transcription," and "Multicellular organismal process." The heatmap in Fig 3 summarizes the relevant enriched GO terms for DEGs in tubers treated with the *S. plymuthica A30* ± *D. solani* at different time points. At early time points, upregulated DEGs were prominently enriched in GO terms related to "responses to stress/stimuli", "cell communication and signaling," "cell wall organization," "cellular homeostasis," and general metabolism, whereas "photosynthetic electron transport chain" was mainly downregulated. Comparison of DEGs at 168 hpi revealed a considerable overlap, both in terms of numbers and gene identity, between +A30−D and +A30+D. Concerning biological processes, more than 20 different GO terms were represented in core sets of DEGs that shared between single and combine treatments, indicating the rather similar nature of the tuber

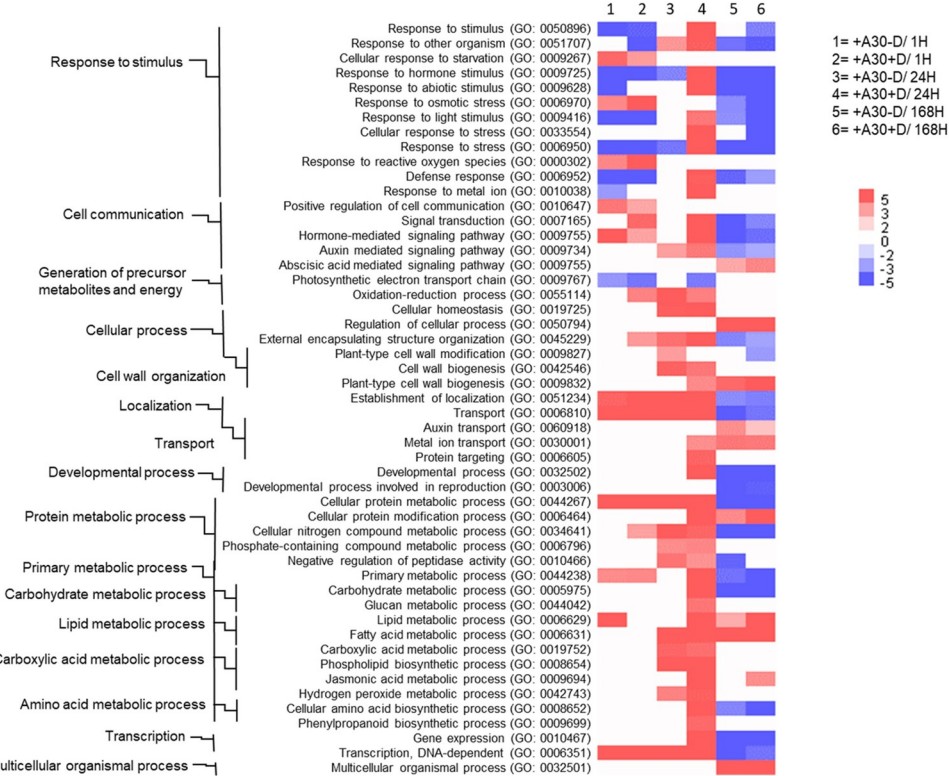

**Fig 3. Overrepresented Gene Ontology (GO) terms within biological processes regulated by *S. plymuthica* A30 before and after challenge with *D. solani* at different time points.** Heatmap represent the p-values of GO terms of up- or down-regulated DEGs (corresponding to DEGs provided in S1 Table). The significance was plotted in red-blue color scale, with white color indicating no significance, red box represents up- and blue box represent down-regulated GO terms. The color intensities indicate level of enrichment score of each GO term.

response to A30 at 168 hpi. GO terms related to "multicellular organismal process", "regulation of cellular processes", "fatty acid metabolic process" and "cellular protein modification process" were enriched within the upregulated genes, while GO related to stimulus and stress response, growth, and development were mainly found in downregulated genes (Fig 3).

Transcriptome profiling of *D. solani*-infected plants with (+A30+D) or without (−A30+D) pretreatment by strain A30 was also investigated at 24 hpi, to allow a further comparison between endophyte-induced ISR and pathogen-activated SAR (S3 Table). In total, 902 and 630 DEGs were identified for −A30+D and +A30+D, respectively. Among these, +A30+D samples exhibited 704 unique upregulated genes, potentially associated with the impact of strain A30 (Fig 4). Functional annotation revealed that +A30+D treatment influenced GO terms "cuticle hydrocarbon biosynthetic process", "oxidation-reduction process", "multi-organism process", "negative regulation of endopeptidase activity" and "response to biotic stimulus" (Fig 4). The top 20 metabolic pathways linked to DEGs in each treatment group are shown in Fig 4. A notable observation is the considerable difference in pathway enrichment categories between the two treatments. By conducting the two RNA-Seq analyses, we concluded that tuber responses

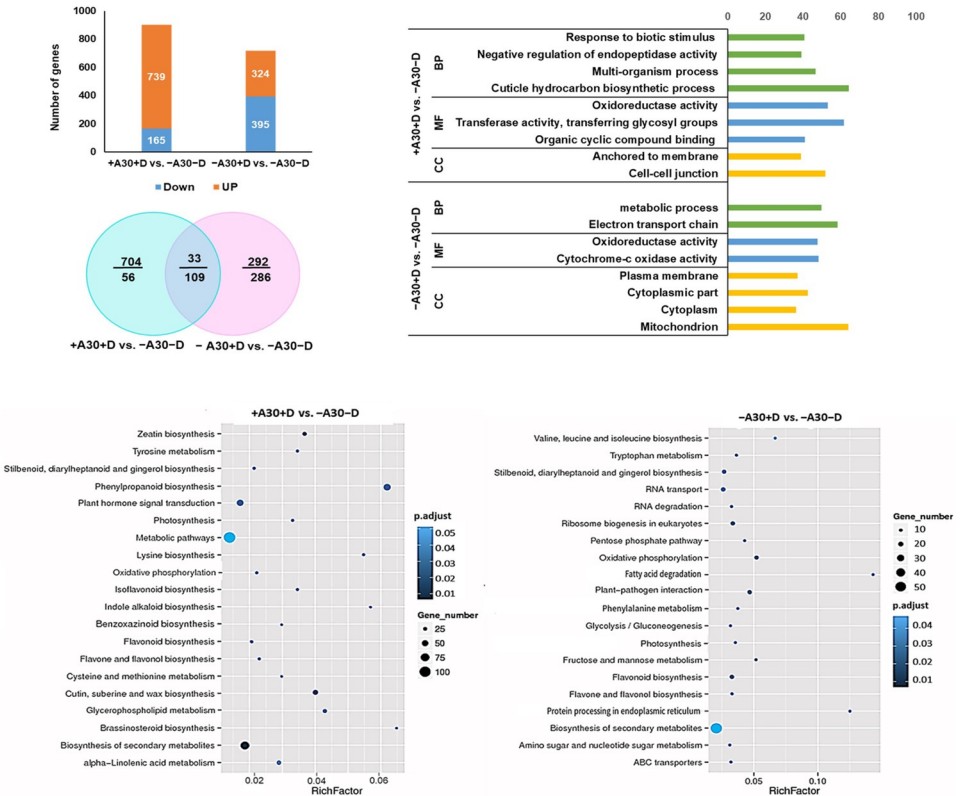

**Fig 4. Transcriptomic changes in potato tubers in response to inoculation with *D. solani* alone or in combination with *S. plymuthica* A30.** The number of up- and down-regulated DEGs in each pairwise comparison of +A30+D and− A30+D against−A30−D (mock control); Venn diagram illustrates the distribution of overlapping and unique DEGs at each comparison; GO enrichment analysis of the DEGs based on three main categories: Biological process (BP), molecular function (MF) and cellular component (CC) for each comparison. The percentage of genes in each category is shown on the top y-axis, while the x-axis displays GO terms within each category; Top 20 KEGG pathway enrichment of DEGs in each comparison, X-axis represents enrichment factor. Y-axis represents pathway name, and the color indicates the p-value. The lower p-value indicates the more significant enrichment. The size of the points indicates the number of DEGs, and the Rich Factor represents the value of the enrichment factor, which is the proportion of the value (the number of DEGs) and the background value (total gene numbers). A larger Rich Factor value indicates more significant enrichment.

to beneficial or pathogenic bacteria can be categorized into five main areas: plant-microbe interaction, response to stress, hormonal regulation, cell wall modification, and primary and secondary metabolism (Fig 3 and S2 Table). This categorization enables the comparison of gene activity modulation between endophytic symbiosis and induced resistance against *D. solani*.

## Transcriptional changes in plant innate immune response

In the "plant-microbe interaction" category, we identified 450 genes encoding a diverse range of receptors, signaling kinases, $Ca^{2+}$ sensors and transporters, G-protein-coupled receptors, inositol phosphates, and ubiquitin ligases (S1 and S2 Tables). Among these, numerous receptor-like kinases (*RLKs*) involved in the perception of bacterial flagellin were identified, including FLS2 receptor-like kinase, brassinosteroid insensitive 1, and somatic embryogenesis receptor-like kinase, which were upregulated at early time points, primarily in +A30+D. Genes encoding L-type and G-type lectin receptor-like kinases (*LecRLKs*), G-protein-coupled receptors (including rac-like GTP-binding protein, ras-related proteins, *GTPases*), and wall-associated kinases (*WAKs*) were activated during the early stages but showed downregulation in the later response. Gene expression profiles indicate the activation of $Ca^{2+}$ signaling, as evidenced by the upregulation of genes associated with $Ca^{2+}$ sensors including calcium-dependent protein kinases, calcineurin B-like proteins, and calcineurin interacting protein kinases. DEGs related to $Ca^{2+}$ influx, including calmodulin-like proteins and calcium-binding proteins, were upregulated in the early time points. Furthermore, genes involved in $Ca^{2+}$ transport, including calcium-transporting ATPases and calcium uptake protein 1, along with those encoding calcium-binding mitochondrial carrier protein *SCaMC1* like providing $Ca^{2+}$ flow from the cytoplasm to mitochondria, were observed. Genes encoding ion channels, including two-pore calcium channel protein 1 and cyclic nucleotide-gated ion channels localized in the plasma membrane for $Ca^{2+}$ efflux to the cytoplasm, were also presented. At early time points, a specific set of genes encoding *MAPKKK* and *WRKYs* with homology to *A. thaliana WRKY22/29* and *WRKY25/33* were mostly induced downstream of receptors to participate in plant disease resistance.

We observed an increase in the expression of genes associated with ubiquitination, well-known as a pivotal "signaling hub" due to their critical role in various signaling events. This included the upregulation of genes encoding E3 ubiquitin ligases, including *PUB21*-like, *PUB23*-like, *PUB24*-like, *ATL27*, *ATL6*-like, *ATL16*-like, *ATL80*-like, and F-box/LRR-repeat proteins. These proteins have been shown to play roles in the early stages of plant defense signaling pathways [33], and their upregulation was notable at the early time points. In results obtained from the first RNA-Seq analysis, DEGs encoding resistance proteins (R) involved in the activation of ETI responses were also upregulated at both 1 and 24 hpi, while they were downregulated at 168 hpi. These included various types of R genes, such as those with toll/interleukin domain including late blight resistance R1-like proteins (TNLs-like) or coiled-coil domain like TMV resistance protein N-like (CNLs-like), as well as disease resistance proteins (NB-ARC domain) homologous to Arabidopsis *RPM1*. Upon closer examination of the second transcriptome profile, it was observed that LRR receptor-like, cysteine-rich receptor-like, and proline-rich receptor kinases were induced, while transcripts of *WAKs* and *LecRLKs* were expressed only in response to +A30+D. Resistance proteins belonging to CNLs-like, TNLs-like, or NB-ARC families were strongly induced in +A30+D conditions (S3 Table).

## Expression pattern of genes associated with stress responses

The expression of genes related to the general response to stimulus or stress was partially activated as an immediate response to A30, but this activation was significantly more pronounced

when the tubers were treated with a combination of A30 and *D. solani* at 24 hpi followed by downregulation at 168 hpi. During the early response, a subset of genes belonging to ROS response were activated in +A30+D but not in tubers treated with single A30 (+A30−D), while others were upregulated in both groups. For example, genes encoding respiratory burst oxidase homolog (RBOH) proteins were upregulated only in +A30+D samples at early time points, whereas these ROS-producing enzymes were downregulated at the later time point (S2 Table). Similarly, an increase in expression of the *RBOHB* gene was noted in tubers infected by *D. solani* (−A30+D) at 24 hpi. Furthermore, a range of ROS scavengers were activated upon A30-induced resistance. However, a stronger antioxidant response was triggered by the combination of pathogen and antagonist, which probably reduced ROS concentration and initiated retrograde ROS signaling to counter oxidative stress. A major portion of genes related to antioxidant systems encoded enzymes involved in glutathione metabolism. Genes for glutathione peroxidase, glutaredoxin, and glutathione S-transferases were predominantly upregulated at early and downregulated at late time point. Other antioxidant enzymes, including peroxidase, superoxide dismutase, peroxiredoxin, thioredoxin-reductase, thioredoxin, polyphenol oxidase, methionine sulfoxide reductase, and heme oxygenase1 were also identified. Interestingly, genes for L-ascorbate oxidase were downregulated in response to both A30 treatments, but upregulated in the case of single *D. solani* inoculation (−A30+D).

The application of A30 exerted regulatory effects on the expression of numerous defense-related genes. At 24 hpi, there was a transient increase in the expression of defense-related genes, including β-1,3-glucanases (*PR2*), endochitinase (*PR3*), thaumatin-like proteins (*PR5*), osmotin-like proteins (*PR5*), the JA signaling marker defensin 1.2, potato antibiotic peptide snakin-1, and germin-like protein involved in thermo and saline tolerance in potato [34]. This heightened gene activity plays a crucial role in inducing systemic resistance and reducing soft rot in tubers colonized by A30. Nevertheless, there was a significant downregulation observed in genes encoding *PR1*, followed by STH-2-like proteins (homologous to *PR10*) at 168 hpi. The beneficial A30 strain also exerted regulatory control over the expression of transcription factors linked to plant stress tolerance, particularly within TF families including *WRKY*, *MYB*, *bHLH* (including *MYC2*), *ERF/AP2*, and heat stress transcription factor (*HSF*), which are involved in both compatibility and resistance processes. *WRKY* genes showed time-dependent differential expression; positive regulators of the JA-mediated pathways, including *WRKY33*, were more expressed at early time points, while those regulated by the SA pathway (including *WRKY3*, *WRKY4*, *WRKY6*, *WRKY30*, and *WRKY48*) [35] were mainly downregulated during the late stages of the interaction.

The transcriptional analysis of 47 genes encoding heat shock proteins (HSPs) revealed a decrease in expression at 168 hpi. These genes mainly included chloroplastic chaperone dnaJ-like and *HtpG*-like, which participate in defense responses. Furthermore, genes encoding HSPs with molecular weights lower than 30 kDa (including *HSP20*) and higher than 70 kDa (including *HSP70*, *HSP83*, and *HSP90*) displayed this regulatory trend. A30 treatments triggered the expression of genes annotated as protease inhibitors (PIs), which are structurally related to various protease families, including aspartic acid PIs, serine PIs, cysteine PIs, metallocarboxy PIs, Kunitz-type PIs, miraculin-like proteins, and two potato type I (*Pin1*) and II (*Pin2*) PIs, commonly known as JA markers [36]. Interestingly, there was a higher number of upregulated PIs in the late response compared to the early response. In the early response, the expression of genes related to proteolytic enzymes, including cathepsin proteinase, aspartic proteinase 1 and 2, nepenthesin1-like, aspartic protease in guard cell-like, and subtilisin-like serine protease were altered, although the majority of them were downregulated in the late response (S2 Table).

## Transcriptional regulation in cell wall modification

Approximately 260 genes were assigned to the functional category of plant cell wall (PCW) modification (S2 Table). Among these, genes encoding cellulose synthase A, cellulose synthase-like, and COBRA-like proteins exhibited upregulation after A30 treatments. Upregulation was also observed for callose synthases responsible for biosynthesis and deposition of callose (β-1,3-glucan) in the early samples, while downregulation of glucan endo-1,3-beta-glucosidase suggested suppression of callose degradation in the later stages. There was no evidence of the activation of callose biosynthesis-related genes during *D. solani* infection (−A30 +D). The largest set of genes within this category was associated with the modification of cross-linking glycans (CLGs), which exhibited upregulation in the early samples and downregulation in the late samples. This gene group encompassed those involved in CLG synthesis, including glycosyltransferases, UDP-glycosyltransferases, xylosyltransferases, glucuronosyltransferases, and β-1,3-galactosyltransferase, which contribute to the biosynthesis of arabinogalactan proteins and the pectic polysaccharide rhamnogalacturonan I [37]. Similar pattern in gene expression was observed for the genes associated with CLG degradation with a fundamental role in cell wall remodeling. These genes encode various hydrolases, including endoglucanases, mannosidases, arabinofuranosidases, xylosidases, and xyloglucan endotransglycosylases/hydrolases (XTHs). XTHs are known to enhance cell wall extensibility during both plant growth and stress responses [38]. Therefore, it is likely that A30 manipulated the host's CLG metabolism, initiating the expansion and remodeling of the cell wall to facilitate host colonization. Interestingly, in contrast to *D. solani* infection (S3 Table), the beneficial strain A30 suppressed host pectin- and rhamnogalacturonan (RG)-degrading enzymes (polygalacturonase, pectate lyase, pectinesterase, and β-galactosidases) (S2 Table). Simultaneously, A30 upregulated genes related to pectin biosynthesis (glucuronosyltransferase) and pectin methyl esterase/pectinesterase inhibitors (PMEIs/PEIs), contributing to the rigidification of the cell wall by increasing the degree of methyl esterification of pectin or potentially inhibiting pathogen pectin methyl esterases [39].

Most of the genes related to plant cell wall proteins, including chitinases, expansins, and three members of the hydroxyproline-rich glycoproteins (HRGPs) family- including arabinogalactan (fasciclin-like), extensin, and proline-rich proteins were upregulated, especially in the +A30+D samples. HRGPs, known for their role as osmoprotectants involved in cell wall lignification and stress tolerance, seemed to remain insoluble in cell walls during both wounding and pathogen attacks [40]. The early increase in expression of two wound-induced defense proteins, *WIN1*-like and *WIN2*, suggests their involvement in callose production during the healing process following pathogen invasion. Similarly, genes related to extensin cross-linking, including extensin-like and leucine-rich repeat extensin-like proteins (*LRX1*, *LRX4*), exhibited induction which was likely involved in strengthening plant cell walls by forming a rigid network. However, the upregulation of genes encoding expansin-like proteins suggests that the endophyte A30 may exploit host expansin as a symbiotic factor, contributing to the loosening of plant cell walls and assisting in plant colonization.

## Changes in plant hormones biosynthesis and signal transduction

Transcriptome profiles and hormonal measurements revealed reprogramming of phytohormone biosynthesis and signaling in tubers treated with either single or combined bacteria (S2 and S3 Tables). Selected DEGs linked to SA, JA, ET, ABA, and auxin are visualized in heatmaps (Fig 5).

Upregulation of several genes related to oxylipin production, specifically in the lipoxygenase (*LOX*) pathway was observed in +A30−D samples. These genes included phospholipase A1

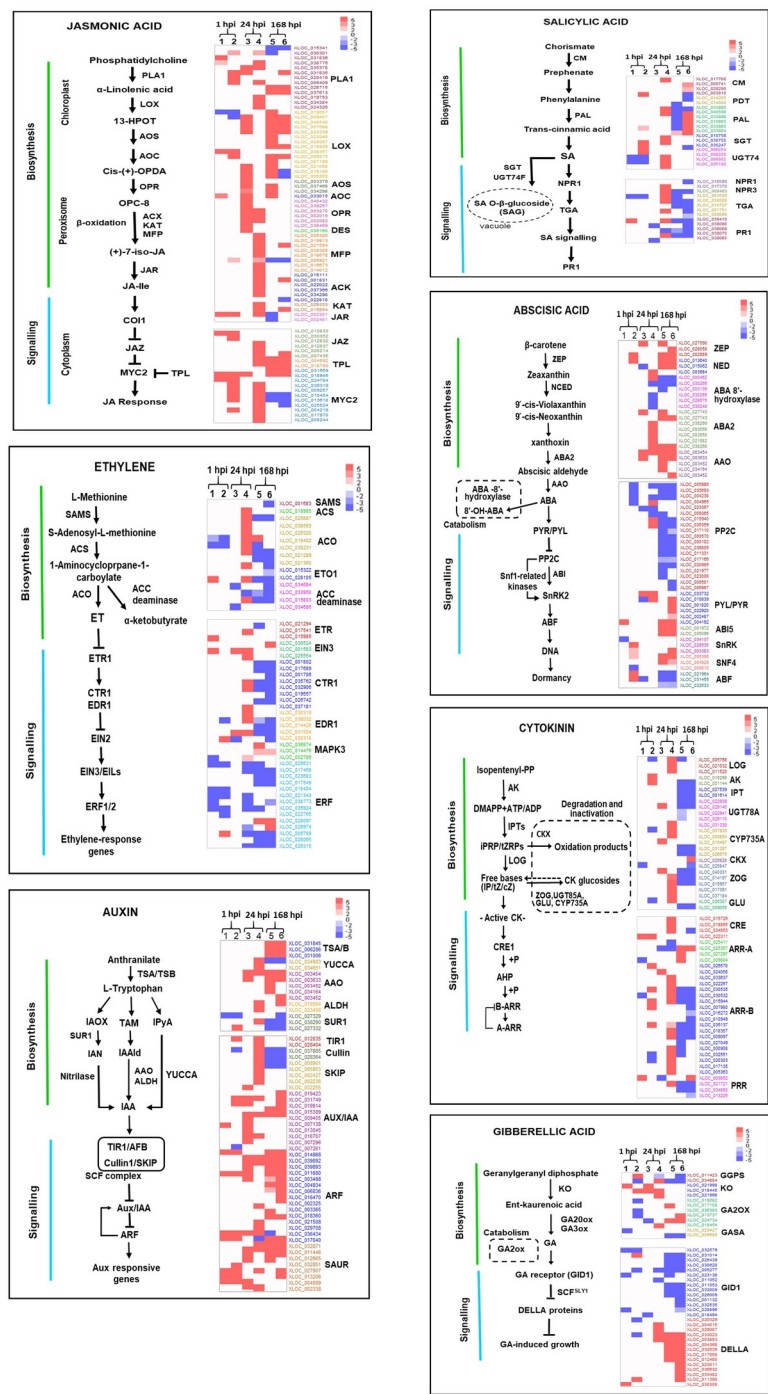

**Fig 5. Heatmap of hormonal system gene expression in potato tubers treated with *S. Plymuthica* A30, with or without *D. solani* Infection.** The columns numbered 1, 3, and 5 represent inoculation with the single strain A30 (+A30-D) at 1 hpi, 24 hpi, and 168 hpi, respectively, while the columns numbered 2, 4, and 6 represent inoculation with both strain A30 and *D. solani* (+A30+D) at the same time points. The vertical axis represents the target genes involved in signaling and biosynthesis pathways, while the horizontal axis represents the expression values in different treatments and time points. Up-regulated genes are depicted in red, while down-regulated genes are shown in blue. The values represent the average expression value (log2 fold change) of each sample in each group. Schematic representations of the biosynthesis pathways were adapted from Simm et al. [41], Lefevere et al. [42], while signaling pathways were adapted and modified from the KEGG pathway "sot04075" of potato.

(*PLA1*), *LOXs* (13S and 9S), allene oxide synthase (*AOS*), allene oxide cyclase (*AOC*), oxophytodienoate reductase, and divinyl ether synthase (*DES*). However, no changes were detected in the gene expression of acyl CoA oxidase (*ACX*), 3-ketoacyl-CoA synthase (*KCS*), 3-ketoacyl-CoA-thiolase (*KAT*), and multifunctional protein AIM1-like (*AIM1*-like), which are involved in the peroxisomal fatty acid β-oxidation machinery of the JA and JA-Ile (jasmonic acid isoleucine) biosynthesis pathway [43]. These findings suggest the production of OPDA but not JA during the single A30 treatment. In contrast, genes in both lipoxygenase and ß-oxidation pathways were induced in the combined treatment (+A30+D) at early time points. The second RNA-seq analysis supported these results for +A30+D samples, showing the induction of DEGs encoding *PLA1*, protein *AIM1*-like, *ACX3*, *KCS11*-like, and *LOXs* at 24 hpi, while no DEGs related to JA biosynthesis were found in tuber samples inoculated with *D. Solani* (−A30 +D) in the early infection phase (S3 Table). Regarding JA signaling, upregulation of genes with close homology to MYC2 was observed during early time points for +A30+D samples, while the expression of JA negative regulators, including jasmonate-zim-domain protein (*JAZ*) from the TIFY family, and protein TOPLESS (*TPL*), was upregulated at later time point. The JAZ repressors, along with their TPL co-repressors, are known to inhibit the activity of *MYC2* [44]. At late time point 168 hpi, *MYC2* activity was likely repressed due to the upregulation of JAZ proteins and TPL in both treatment groups, +A30−D and +A30+D (Fig 5), suggesting the deactivation of the JA signaling as a late response to A30 treatment. Hormonal measurements were consistent with the transcriptome analysis, revealing increased OPDA levels in both early and late response to A30, most notably in +A30−D samples. JA content increased at 24 hpi under combined treatment, while only a slight increase was observed in the samples treated with single A30 treatment. In contrast, a high accumulation of JA was detected during *D. solani* (−A30+D) infection at 168 hpi (Fig 6).

Following A30 treatments, only some genes related to the synthesis of SA precursors were moderately upregulated, including phenylalanine ammonia lyases (*PAL*), chorismate mutase, chorismate synthase and 4-coumarate: CoA ligase-like, which can be interconnected in other metabolic pathways (phenylpropanoid and prephenate pathways). Several transcripts encoding pathogen-inducible salicylic acid glucosyltransferase (*UGT74F1* and *UGT74F2*) that convert SA to inactive SAG [45], were upregulated in +A30+D samples at 1 and 24 hpi. Although transcripts encoding the markers of SA pathways, including pathogenesis-related protein 1 (*PR1*)-like, SA receptor *NPR5*, *NPR3*, *NIM1*-like protein 2, and DNA-binding proteins TGA (including *TGA1* and *TGA2.1*-like) exhibited transient upregulation at 24 hpi, while most of them were downregulated at 168 hpi. The second transcriptome profile (S3 Table) revealed induction of the SA pathway in *D. solani* inoculated tubers (−A30+D) with upregulation of DEGs encoding *PAL*, *PR1-A1* and transcription factor *TGA1*-like. Measurement of hormone production revealed that SA was the main hormone that was increased upon *D. solani* inoculation (−A30+D) within 24 hpi (47.8 ng/g FW), and then its level decreased at 168 hpi (14.7 ng/g FW). As shown in Fig 6, the SA level remained unchanged in both +A30−D and +A30+D samples. In addition, the levels of SAG in the A30-treated tubers increased and reached about 2-fold higher levels in both treatments at 168 hpi.

The expression profiles in tubers upon A30 treatment indicated the upregulation of ET biosynthesis-related genes, including ACC oxidase, ACC synthase, and S-adenosylmethionine synthase 2-like (*SAMS2*-like), along with the downregulation of ET overproducer-like 1 (*ETO1*), a negative regulator of ACC synthase (Fig 5). Interestingly, genes encoding ACC deaminase and its homolog D-cysteine desulfiderase [46] displayed different expression patterns during the early and late response. Upregulation of ACC deaminase in the early time points most likely reduced the ET production [47], which could alleviate ET stress possibly caused by the perception of both microbes. However, the downregulation of these genes in the

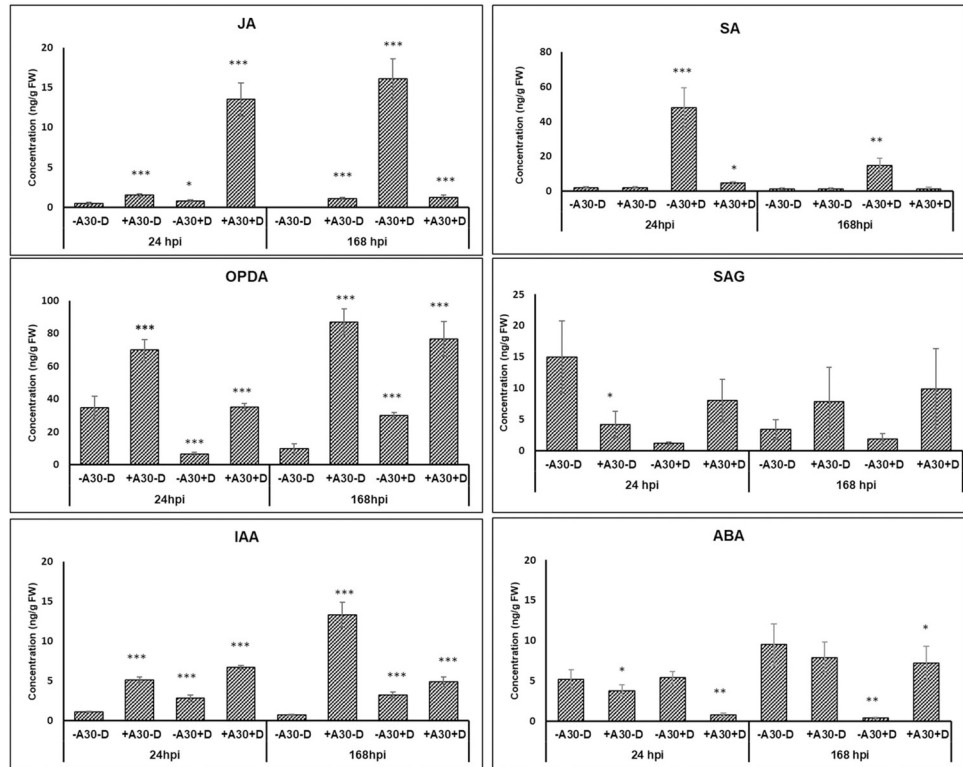

**Fig 6. Changes in phytohormone content of potato tubers treated with single *S. plymuthica* A30 (+A30–D), single *D. solani* (–A30+D), combination of both bacterial strains (+A30+D), and control (-A30-D) at 24 hpi and 168 hpi.** Metabolite concentration (ng/g fresh weight) were measured for SA, JA, OPDA, SAG, ABA, and IAA. Error Bars represent means of five replicates ± SE. Stars indicate significant differences at p ≤ 0.05 (*), p ≤ 0.01 (**) and p ≤ 0.001 (***).

late response 168 hpi may restore the ET content to the normal level after stress relief and when the host plant recognizes endophyte A30 as a beneficial microbe. Transcripts linked to ET perception and signaling including ET receptor 1 (*ETR1*), *ETR2*, constitutive triple response 1 (*CTR1*), and serine/threonine-protein kinase enhanced disease resistance 1 (*EDR1*) were induced in the early samples of +A30+D treatment, while repressed at the late time points of all treatment groups. In Arabidopsis, *CTR1* and *EDR1* are Raf-like protein kinase members that regulate ET signal transduction by inactivating *EIN3* and downstream *ERF* transcription factors [48]. Interestingly, plenty of ET-responsive transcription factors (*ERFs*) including *ERF1*, *ERF2*, *ERF3*, *ERF4*, *RAP2*, and *AIL5*-like were found in a downregulated list after A30 application (S2 Table), whereas *ERF1*-like and *ERF2*-like were upregulated under *D. solani* infection (S3 Table). ET-associated immunity together with the ERF-branch of the JA pathway is typically activated against pathogens with necrotrophic and hemibiotrophic lifestyles [49].

Genes involved in ABA biosynthesis including 9-cis-epoxy carotenoid dioxygenase, zeaxantin epoxidase, violaxanthin de-epoxidase, xanthoxin dehydrogenase (*ABA2*) and abscisic-aldehyde oxidase (*AAO*) were upregulated for both +A30–D and +A30+D samples at early and late time-points (Fig 5). It was along with clear downregulation of genes annotated as ABA 8'-hydroxylase (*CYP707A*), the key gene for ABA catabolism, which controls the seed germination in Arabidopsis [50]. Likewise, A30 treatments resulted in the accumulation of ABA over time points in both treatments +A30–D (from 3.8 ng/g FW at 24 hpi up to 7.9 ng/g FW at 168 hpi) and +A30+D (from 0.8 ng/g FW at 24 hpi up to 7.2 ng/g FW at 168 hpi). Potato tubers

infected by single *D. solani* (–A30+D) showed an obvious decrease in ABA concentration from 24 hpi (5.4 ng/g FW) to 168 hpi (0.4 ng/g FW) (Fig 6).

Transcriptome profiles displayed downregulation of genes encoding probable phosphatase 2C (*PP2C*), a negative regulator in ABA signaling, and upregulation of genes annotated as ABA receptor PYR/PYLs, sucrose nonfermenting 1 (SNF1)-related protein kinases (*SnRKs*) and ABA-insensitive 5 protein-like (*ABI5*), suggesting stimulation of ABA signaling in response to A30. Furthermore, genes with function in ABA-dependent stress response such as ABA-responsive element binding factors (*ABFs*) regulating osmotic stresses in potato [51], ABA negative regulator *ABR1*, dehydration-induced protein *ERD15* acting as ABA negative regulator in drought and freezing tolerance [52], dehydration-induced protein 19, dehydration-responsive element-binding protein 2C, *DREB1*, and ABA stress ripening in the regulation of osmotic, salinity and cold stresses [53] were upregulated at early and downregulated at the late response (S2 Table). Our data also show changes in the expression level of several genes involved in ABA-mediated dormancy processes. Among them, the Mother of FT and TFL 1 (*MFT1*) as promoters of seed dormancy through the regulation of ABA, OPDA, and GA [54], long hypocotyls 5 (*HY5*) in activation of ABI5 expression, ABA light signaling pathways and inhibition of seed germination [55], protein FRIGIDA that delays flowering by increasing the expression of floral repressor FLC (flowering locus C) via ABA and GA cross-talk [56], dormancy-associated *MADS-box* factors with an essential role in bud endodormancy via induction of ABA biosynthesis [57], *HVA22* as an ABA/stress-induced protein with a role in the maintenance of seed dormancy [58], arm repeat protein interacting with abf2 (*ARIA*), a positive regulator of ABA response with role in negative regulation of seed germination [59] and *WRKY41* as a positive regulator of ABI3 expression during seed dormancy [60] were found to be upregulated at 168 hpi.

The hormonal analysis revealed an increase in IAA level in response to both bacterial strains, albeit the increase was notably more pronounced in the presence of A30 compared to the pathogen (Fig 6). The upregulation of genes involved in IAA biosynthesis indicates the activation of auxin production in tubers treated with A30. Additionally, numerous genes involved in auxin-responsive pathways (including Aux/IAA, ARF, and SAUR) were detected at all-time points (Fig 5). Other genes encoding IAA-amido synthetase GH3, transport inhibitor response 1 (*TIR1*), SCF ubiquitin ligase complex F-box protein SKIP-like, and cullin (cullin 1, cullin 3), involved in proteasome-mediated degradation of AUX/IAA, were identified, with most of them showing induction at early time points in the presence of both the pathogen and A30. Additionally, genes involved in polar auxin transport, including ATP-binding cassette transporter subfamily B (*ABCBs*), auxin efflux facilitator SlPIN4 (*PIN4*), nucleoside diphosphate kinase 2, calcium-binding protein PBP1, and auxin resistant1/like auxin resistant (AUX1/LAX), were induced at 24 hpi and decreased at 168 hpi. IAA is mainly synthesized via the tryptophan aminotransferase of the Arabidopsis/YUCCA biosynthetic pathway [61]. In our findings, the upregulation of genes encoding anthranilate synthase, tryptophan synthase, YUCCA family of flavin monooxygenases 1 (*YUCCA1*), indole-3-acetaldehyde oxidase, aldehyde dehydrogenase NAD$^+$ and IAA-amino acid hydrolase ILR1-like 1, involved in releasing active auxin from its amino acid conjugate [62], were detected in both early and late response, indicating an increase in auxin biosynthesis under A30 colonization. Meanwhile, SUR2 (cytochrome *P450 CYP83B1*) converting auxin precursor indole-3-acetaldoxime to indole-3-thiohydroximate, was upregulated at 24 hpi (Fig 5).

Transcriptome profiling related to CKs biosynthesis presented opposite expression direction in early and late response to A30 (S2 Table). Upregulation of genes linked to isopentenyl adenosine-5'-monophosphate -dependent pathway at 24 hpi suggests transit accumulation of CK, followed by a decrease of gene expression at 168 hpi. The two primary members of this

pathway are adenylate-isopentenyl transferase 5, observed in the second transcriptome profile, and cytokinin riboside 5'-monophosphate phosphoribohydrolase, detected in the first transcriptome profile [63]. CK homeostasis is affected by several key pathway enzymes for hormone level maintenance. Among these regulatory enzymes, genes for side-chain modification of CKs structures (cytokinin trans-hydroxylase (*CYP735A*)), N-glucosylation (UDP-glycosyltransferase 85A (*UGT85A1*, *UGT85A2*), O-glucosylation (zeatin O-glucosyltransferase (*ZOG*)) and hydrolysis of O-glucosides (β-glucosidase) [64] were found among upregulated genes at early response 24 hpi, while they were subsequently downregulated at 168 hpi in both –A30+D and +A30+D. The expression of a type-B response regulator (*ARR-B*) as a positive regulator was induced, while a type-A response regulator (*ARR-A*) expression as a negative regulator of CK was reduced. The *ARR-A* was considered as negative-feedback regulators of CK signaling in response to stress at early time points that showed a reverse expression trend at 168 hpi. Additionally, cytokinin response factor *CRF4*, a member of the ERF in both cytokinin and ET -activated signaling pathways, was found upregulated at 1 hpi. Interaction with strain A30 induced gibberellin-inactivating enzymes (gibberellin 2-beta-dioxygenases-like encoded by *GA2ox8* and *GA2ox1*) at 168 hpi, suggesting a decrease in GA content as the late response. This feedback mechanism also appears to operate at the level of GA perception with massive downregulation of gibberellin receptor *GID1B* and members of carboxylesterase with similar *GID1* function in both early and late time points, along with upregulation of GA negative regulator DELLA protein (*RGL2*-like proteins) and GRAS family protein SCL-like (scarecrow-like) at 168 hpi. Overexpression of DELLA mediates the GA-DELLA signaling and promotes the expression of *AB15* to inhibit GA-dependent processes [65]. Additionally, we found at 168 hpi repression of a gene annotated as GA-inducible *GAMYB* that encodes an *R2R3-MYB* transcription factor responsible for flowering and sugar metabolism including α-amylase during germination [66]. In the second transcriptome analysis, GA biosynthesis and responsive genes induced upon *D. solani* infection by the upregulation of ent-copalyl diphosphate synthase, farnesyl-diphosphate farnesyltransferase, *GA20ox*, and *GA3ox4*, along with downregulation of DELLA protein *RGL2* and *SCL15*, suggest an important role for GA in the response of potato tubers to *D. solani* infection (S2 Table).

## Alteration in gene regulation associated to primary and secondary metabolisms

Following A30 treatment, most of the DEGs associated with the "photosynthetic electron transport chain" were downregulated, although exceptions were noted in +A30+D samples at 24 hpi (S2 Table). Notably, an early induction of genes involved in oxidative phosphorylation, including NADH: ubiquinone oxidoreductase and NADH dehydrogenase (mitochondrial complex I), succinate dehydrogenase (complex II), cytochrome c oxidase (mitochondrial complex IV), cytochrome c reductase (complex III), and proton-transporting ATP synthase (such as F-type H+ transport ATPases), was exclusively observed in +A30+D samples, suggesting a modification in the mitochondrial electron transport chain (ETC) and energy utilization upon A30 treatment.

Co-inoculation of endophytes with pathogens led to an enhancement of aerobic respiration in plant carbon metabolism, resulting in the upregulation of various DEGs such as fructokinase, gluconokinase, fructose-bisphosphate aldolase, enolase, and phosphoenolpyruvate carboxykinase at 24 hpi. Additionally, genes involved in acetyl-CoA biosynthesis and oxaloacetate regeneration, such as malate dehydrogenase and pyruvate dehydrogenase, were upregulated, whereas those encoding alcohol dehydrogenases and lactate dehydrogenases associated with anaerobic respiration were either repressed or not expressed. Genes related to the systemic

transport of carbohydrates, including invertase, glucose transporter, SWEET bidirectional sugar transporter, and protein phloem protein, as well as a phosphate transporter PHO1, exhibited partial upregulation during the early time point but were significantly downregulated in the late response. Moreover, the induction of five DEGs annotated as nitrate transporters, which play a role in balancing carbon and nitrogen metabolism, was observed after A30 inoculation.

Most genes involved in fatty acid metabolism, such as fatty-acyl-CoA synthase, acetyl-CoA carboxylase, and acetyl-CoA carboxylase/biotin carboxyl carrier protein, displayed upregulation. Concurrently, DEGs encoding phospholipases (PLA1 and PLD) and galactolipids monogalactosyldiacylglycerol (MGDG) and digalactosyldiacylglycerol (DGDG), as well as genes encoding fatty acid desaturases, including omega-6 fatty acid desaturase and delta8-fatty-acid desaturase, involved in polyunsaturated fatty acid (PUFAs) biosynthesis, were upregulated upon A30 treatment, which release fatty acids that serve as substrates for the lipoxygenase pathway. Early induction of genes participating in the fatty acid beta-oxidation pathway and JA biosynthesis, such as ACX, peroxisomal AIM1-like prote in, and bifunctional enzyme enoyl-CoA hydratase/3-hydroxyacyl-CoA, was specifically observed in combined treatment.

Wax is a complex mixture of very long-chain fatty acids (VLCFA). In our data, genes related to VLCFA formation, including fatty acyl-ACP thioesterase A (*FATA*), malonyl-CoA-acyl carrier protein, long-chain acyl-CoA synthetase (*LACS*) for esterification of VLCFA to acyl-Coenzyme A, and acyl-coenzyme A thioesterase [67], were detected. Fatty acid Acyl-CoA elongation is the beginning point for wax and suberin biosynthesis, which is a complex process involving several enzymes, such as 3-ketoacyl-CoA synthase (*KCS*), 3-oxoacyl-[ACP] reductase (*KAR*), very-long-chain 3-hydroxyacyl-CoA dehydratase (*PASTICCINO*2-like), enoyl-ACP reductase (*ENR*) and very-long-chain enoyl-CoA reductase (*ECR*), which were induced at early and diminished later on. Furthermore, key genes associated with the alcohol-forming pathway of the wax mixture, such as wax-ester synthase/diacylglycerol O-acyltransferase and aldehyde decarboxylase eceriferum 1-like (*CER1*-like) [68], were primarily activated for +A30 +D at early time points.

The application of endophyte A30 stimulated the metabolism of various amino acids (S2 and S3 Tables). This was evident from the upregulation of genes encoding PALs, phenylalanine-tRNA ligases, and tryptophan synthase, may signify the induction of biosynthesis pathways for tryptophan and phenylalanine. Early induction of cysteine synthases and cysteine desulfurases involved in Cys biosynthesis and degradation, respectively, probably control the level of free cysteine cytotoxicity in the cell. The activation of the methionine salvage pathway after +A30+D inoculation in the early stages, which includes genes coding for S-adenosyl-methionine synthase 2-like (*SAM2*-like), peptide methionine sulfoxide reductase, and S-adenosylmethionine decarboxylase proenzyme (*AdoMetDC*), might be associated with the sequential stages of ET biosynthesis. Furthermore, there was an increase in the expression of genes related to the synthesis of arginine (Arg) and asparagine (Asp), which serve as primary nitrogen storage compounds and influence various physiological processes in the plant. This included the upregulation of genes such as arginine N-methyltransferase, asparagine synthetase, and aspartate aminotransferase. Additionally, genes involved in the conversion of Arg to polyamine agmatine (arginine decarboxylase and agmatine deiminase-like), ornithine to polyamine putrescine (ornithine decarboxylase), and ornithine to non-protein amino acid citrulline (ornithine carbamoyltransferase), were upregulated at early time points. These findings suggest a coordinated response in polyamine production, highlighting the potential role of A30 in modulating plant stress responses. In the early samples, the genes encoding pyrroline-5-carboxylate, involved in the conversion of glutamate to proline, as well as the proline transporter and glutamate-glyoxylate aminotransferase 2 (GGT2), which serves as a homolog of Ala

aminotransferase, exhibited significant upregulation. The peroxisomal GGT2 enzyme facilitates the conversion of glyoxylate to glycine and pyruvate, providing substrates for continuous energy production, fatty acid β-oxidation, and lipid mobilization [69].

The transcriptome analysis uncovered alterations in secondary metabolite pathways in tubers subjected to A30 treatment, encompassing pathways such as "Phenylpropanoid biosynthesis," "Zeatin biosynthesis," "Cutin, suberin, and wax biosynthesis," "Flavonoid biosynthesis," "Stilbenoid, diarylheptanoid, and gingerol biosynthesis," "alpha-Linolenic acid metabolism," and "Benzoxazinoid biosynthesis" (Fig 3 and S2 Table). The largest subcategory was phenylpropanoid biosynthesis, showcasing upregulated DEGs involved in lignin biosynthesis. Genes encoding enzymes for monolignol biosynthesis, such as cinnamyl-alcohol dehydrogenase, ferulate-5-hydroxylase, caffeoyl-CoA O-methyltransferase, cinnamoyl-CoA reductase, coniferyl-alcohol glucosyltransferase, and 2-alkenal reductase, were induced in A30-treated tubers, particularly following pathogen infection. Additionally, oxidative enzymes potentially involved in the polymerization of monolignols, including laccases (laccase-14-like and laccase-17-like) and lignin peroxidase (lignin-forming anionic peroxidase-like), as well as beta-glucosidases facilitating the transport and storage of monolignols, exhibited pronounced upregulation.

Similarly, various genes involved in suberin biosynthesis were predominantly induced at early stages and to a lesser extent the late time point (S2 Table). Upregulation of genes responsible for the core phenylpropanoid pathway, which catalyzes the formation of hydroxycinnamate structures such as p-coumaric acid, caffeic acid, ferulic acid, and sinapic acid involved in suberin-phenolic monomer biosynthesis, was observed in +A30+D samples. Additionally, the polymerization of phenolic domains was supported by the upregulation of several genes encoding peroxidases, including suberization-associated anionic peroxidase, cationic peroxidase 1-like, and other cell wall peroxidases. Genes related to the biosynthesis of aliphatic monomers, which are also detected in wax biosynthesis, such as KCS, ECR, PASTICCINO2-like, and fatty acid omega-hydroxylases (CYP86A, CYP86B, and CYP94A), were found to be upregulated. Genes with function in esterification, deposition, and assembly of aliphatic domains, including glycerol-3-phosphate acyltransferase and GDSL-type esterase/lipase proteins acting as "suberin synthase" proteins, as well as omega-hydroxypalmitate O-feruloyl transferase, involved in the accumulation of the ferulate constituent of suberin, showed induction at early stages and repression at later stages. Furthermore, several MYBs and WRKYs (including WRKY33 and WRKY1) implicated in the regulation of suberin biosynthesis, and the ABC transporter G family [70], known to transport suberin monomers to the apoplastic space, were mainly induced.

In +A30+D samples at 24 hpi, expression of genes encoding enzymes of the flavonoid pathway exhibited upregulated, including chalcone synthase and flavonol synthase/flavonol 3 hydroxylase, as well as those involved in the anthocyanin pathway including dihydroflavonol reductase and leucoanthocyanidin dioxygenase-like enzymes. Additionally, genes responsible for the formation of dihydroquercetin and cyanidin 3-O-glucoside, both potent plant antioxidants, were upregulated. Moreover, genes related to isoflavonoid-derived antimicrobial compounds, such as isoflavone 2'-hydroxylase-like and isoflavone reductase, showed increased expression in +A30+D samples at 24 hpi. There was also an increase in the expression of genes associated with terpenoid backbone and sesquiterpene biosynthesis, including farnesyl diphosphate synthase and geraniol 8-hydroxylase-like enzymes. Interestingly, the downregulation of diterpenoid enzymes such as geranygeranyl pyrophosphate synthase and ent-kaurene oxidase at the late time point correlated with the repression of gibberellin-related genes, indicating their role as precursors for this phytohormone. Early induction of vetispiradiene synthase which plays a role in the biosynthesis of various sesquiterpenoid phytoalexins was observed in both single and combined treatments. The genes encoding cytochrome P450, including three

*CYP71D7* and one *CYP71D55*, annotated to code for premnaspirodiene oxygenase-like proteins that are responsible for solavetivone and rishitin biosynthesis [71] were upregulated during the early response, while downregulated at the late time point. In potato tubers and tomato fruits, rishitin, solavetivone, and vetispiradiene are recognized as major sesquiterpenoid phytoalexins [15]. Additionally, three genes encoding trans-resveratrol di-O-methyltransferase-like enzymes, responsible for catalyzing the phytoalexin pterostilbene from resveratrol, and two genes encoding neomenthol dehydrogenase-like enzymes involved in the production of neomenthol with antimicrobial activity [72], were found to be upregulated in +A30+D samples at 24 hpi. We also observed a rise in the expression of genes involved in the biosynthesis of Benzoxazinoid and their derivatives, which are effective in defence and allelopathy, including 2,4-dihydroxy-1,4-benzoxazin-3-one (DIBOA), 2,4-dihydroxy-7-methoxy-2H-1,4-benzoxazin-3(4H)-one (DIMBOA), and DIBOA-Glc dioxygenase (BX6). Interestingly, a reduction of genes connected to steroid glycoalkaloid biosynthesis, namely α-solanine and α-chaconine with role in induction of tuber greening, was observed at 168hpi after transit upsurge at 24 hpi.

### Validation of selected DEGs in RNA-Seq data by real time-PCR (RT-PCR)

To verify the time course RNA-Seq data, qRT-PCR analysis was conducted on 34 identified genes (S4 Table) across the time course in either one or both treatments of +A30-D and +A30+D. The expression profiles obtained from qRT-PCR analyses closely matched those from the RNA-Seq data (Fig 7). Pearson's correlation coefficients were calculated to assess the similarity in expression trends between the two experiments, yielding a significant correlation (r2 = 0.7182; Fig 7), thus confirming the accuracy of our RNA-Seq data. To confirm the second transcriptome profile, qRT-PCR analysis was conducted on 15 selected DEGs (S4 Table). The expression patterns of these chosen genes exhibited consistency between RNA-Seq and qRT-PCR analyses (Fig 7), albeit with slight variations in the fold change (FC) values. This suggests that the two techniques might have different levels of sensitivity particularly in distinguishing members of multigene families and orthologous genes. Nonetheless, the correlation coefficients for -A30+D and +A30+D were 0.67 and 0.69, respectively (Fig 7), indicating good consistency between the two analysis techniques and further supporting the reliability of the RNA-Seq analysis.

## Discussion

In this study, the interaction between potato tubers, biocontrol agent *S. plymuthica* A30 and pathogen *D. solani* have been analyzed from the perspective of potato response to biocontrol agent and induced resistance against soft rot disease. Our prior investigation demonstrated the significant reduction in *D. solani* DNA levels on A30-treated tubers, thereby confirming the direct antagonism on pathogen growth [12]. Here, we discuss data suggesting that the enhanced resistance observed in A30 bacterized tubers is not only linked to a direct antagonism but is also associated with the induction of structural and biochemical barriers within the host tissues.

### Potato tubers exhibit distinct expression profiles in the early and late response to *S. plymuthica* A30

In the early phase of interaction, strain A30 exhibited a dual effect depending on the presence or absence of the pathogen. The application of single strain A30 had a minimal effect on host transcript levels when the pathogen was absent. This reflects the behavior in other endophytic bacteria, which typically do not require additional host plant resources in the absence of pathogens [73–75]. However, upon treatment of potato tubers with A30 in combination with *D*.

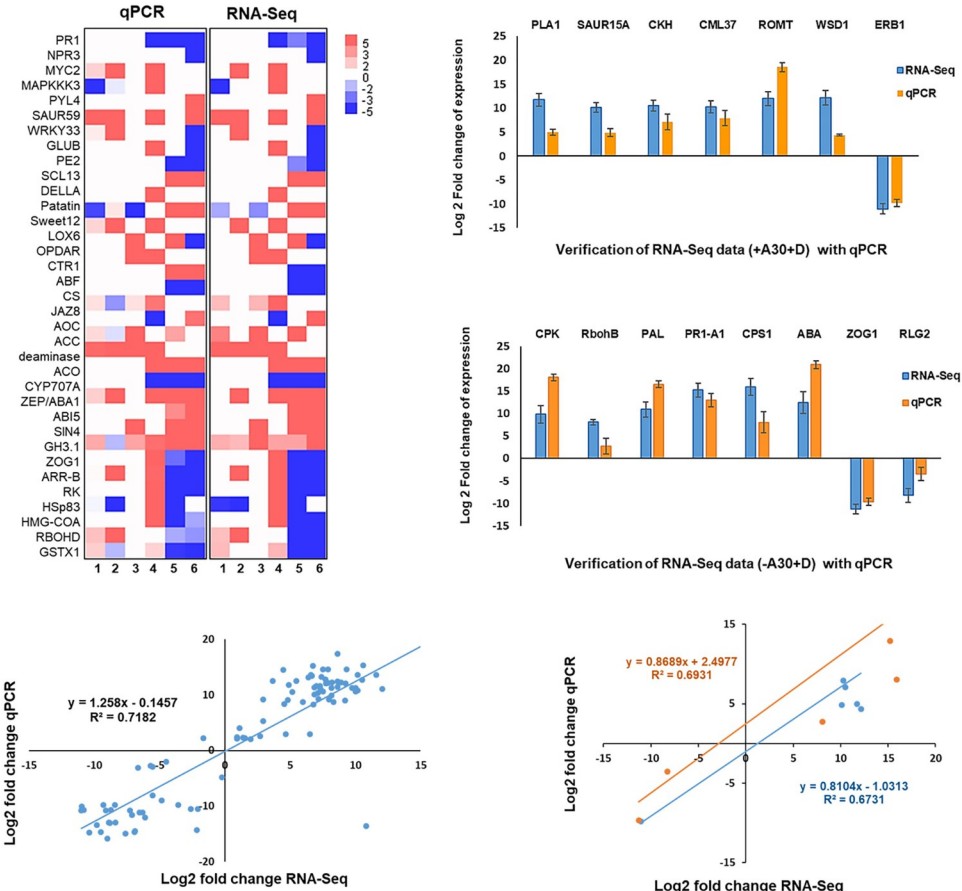

**Fig 7. Validation of RNA-Seq analysis with qRT-PCR assay.** Heatmap represented comparisons of expression profiles for selected DEGs between time course RNA-Seq and qPCR. The vertical axis labels the target genes examined, while the horizontal axis simplifies the sample group identities as follows: (1) +A30-D/1h; (2) +A30+D/1h; (3) +A30-D/24h; (4) +A30+D/24h; (5) +A30-D/168h; (6) +A30+D/168h. Bar graphs presenting the expression values determined by qRT-PCR for selected DEGs from the second RNA-Seq in potato tubers inoculated with a single *D. solani* and with combination of strains *D. solani* and *S. plymuthica* A30. Regression plot displaying the direct correlations between time course RNA-Seq and qRT–PCR data for each DEG. Statistical analysis was conducted using coefficient correlation analysis by the Pearson test (p < 0.05). Trend lines depicting qRT-PCR validation of the second RNA-Seq for -A30+D and +A30+D. Statistical analysis was performed using coefficient correlation analysis by the Pearson test (p < 0.05). In all experiments, qPCR data were obtained from three independent cDNA sets from three independent experiments, normalized to eukaryotic elongation factor 5A3, and expressed as means of log2 (ΔΔCt) ± SEM (standard error of the mean).

*solani*, a rapid and more intense transcriptional reprogramming occurred, indicating a prioritized defense mechanism against pathogenic infection. In the late stage of interaction, the transcriptional changes in tuber response to A30 showed significant overlap in terms of affected genes and their expression patterns, regardless of whether the pathogen was present or absent, which suggests a similar effect of A30 in both single and combined treatments.

During the early response, we observed both analogy and convergence among genes linked to the signal transduction category, particularly those affected during ISR or SAR. One potential explanation is that these alterations signify the shared events within the plant's recognition system, highlighting the detection of common structural features present in both beneficial and pathogenic bacterial strains. Nonetheless, the recognition of diverse microbial ligands may facilitate reaching the signaling threshold required for the activation of innate immunity.

Thus, the perception of both bacterial strains, P/MAMPs, ultimately triggers a more robust activation of immune responses during the initial phase of interaction. In addition, the results suggest that endophyte A30 might employ tactics to enable the host to differentiate it from pathogens. For example, certain *LecRLKs* including *LecRK-IX.2* and *LecRLK1*, exhibit specific involvement in response to A30, which could be supposed as host symbiotic receptors that initiate symbiosis signaling. The *LecRLKs* have been identified as important players in the plant-microbe symbiotic relationship including legume-rhizobia symbiosis [76] as well as the mutualistic symbiotic associations with mycorrhizal fungi [77].

Similar to the known beneficial effects of other endophytic bacteria, the ISR triggered by strain A30 involves an increase in the activation of both MTI and ETI immune responses, which is accompanied by the stimulation and initiation of downstream defense signaling cascades, including oxidative burst, $Ca^{2+}$ influx, protein kinase cascades, and induction of various R genes during the initial phase of interaction. The magnitude of this response significantly increases when confronted with *D. solani*. The MTI and ETI responses rendered A30-treated tubers more responsive to subsequent *D. solani* infection, prompting speculation that signals from the pathogen are essential for the accumulation of defense products and accelerate endophyte performance to evoke a fully-fledged ISR response. ROS generation occurs rapidly in the early response to both pathogen infection and strain A30 symbiotic interaction, indicating that ROS production is a hallmark of successful recognition. In the absence of the pathogen, this reaction appears to be mild in response to single strain A30 due to less abundant gene expression, which likely results in a low level of ROS production. This non-toxic level of ROS functions as both a local and long-distance signal for host defense priming against subsequent pathogen challenge. In the cases of combined treatment, the expression of *RBOH* and other regulatory genes in ROS signaling was significantly induced, probably leading to a rapid accumulation of apoplastic ROS and enhancing the ROS-dependent defense mechanisms. In contrast to pathogen-induced PTI, which often results in significant cellular damage, the early transcriptional reprogramming of tubers in response to individual A30 was transient and relatively moderate, which may be attributed to A30's capacity to manipulate plant immune mechanisms, facilitating the establishment of a mutually beneficial relationship with the host.

The beneficial endophyte A30 needs to establish sophisticated communication with its host, enabling colonization while evading the host's MAMP-triggered immunity response. This phenomenon was observed during the later stages of interaction with A30, wherein the downregulation of genes associated with plant immunity and signaling implies the suppression of defense responses initiated during earlier stages. Many signaling components, plant receptors, disease resistance proteins, and defense related genes, including *PR1*, *MYBs*, and *WRKYs* (including *WRKY33* and *WRKY70*) were repressed, representing a weakened recognition of M/PAMPs, which is required to establish the intimate associations with plants. Concurrently, the late upregulation of three members of the plant U-box family of ubiquitin E3 ligases (*PUB21-*, *PUB23-* and *PUB24*-like), involved in the proteasomal degradation of PRR complexes for the negative regulation of MTI [78], introduces an additional layer of complexity. Due to a weakened signaling network in the late phase of interaction, genes encoding *RBOHs* and $Ca^{2+}$ signaling tended to be downregulated. Thus, the reduction of the respiratory burst, along with the decreased expression of heat shock proteins (including HSP90), and endoplasmic reticulum stress components (including syntaxin 121), suggest A30's ability to either prevent continuous stimulation or better tolerance of stresses without induction of oxidative stress in the plant. In addition, the decrease in MAMP-triggered ROS burst was linked to the activation of different ROS scavengers within the thioredoxin-glutaredoxin system, peroxidases, and polyphenol oxidase, displaying varied patterns in both combined and single treatments. Similar trends of decreased host immune responses have been recorded in several

biocontrol bacteria, including *B. subtilis* FB17 [79], *P. fluorescens* WCS417r [80], and *B. velezensis* F21 [81]. Despite this, the apparent negative impact on general plant immune responses did not lead to an increase in host susceptibility to pathogen and infection threats. The absence of soft rot symptoms in the late samples may primarily rely on direct antibiosis of strain A30 against pathogens or stimulation of the plant in the production of antimicrobial metabolites, including proteinase inhibitors and phenolic compounds. On the other hand, some ISR-inducing endophytes have evolved robust tolerance to antimicrobial compounds generated by the plant immune response [82]. Proteinase inhibitors are the predominant group of potato storage proteins with antimicrobial properties that hold a significant role in the wound-induced defense response. Additionally, proteases are also essential enzymes for plant defense which serve various functions ranging from pathogen recognition to the initiation of plant hypersensitive response [83]. Both beneficial microbes and pathogens could manipulate the plant proteinase inhibitors to inhibit plant defense proteases to ensure colonization of their host. For instance, the symbiotic fungus *Epichloë festucae* in association with grasses inhibits apoplastic plant proteases to overcome plant immunity and effectively colonize the host [84]. In our data, the induction of PI-protease complex at the early response to the combination of A30 and *D. solani* suggests their role in plant immunity response. Conversely, the downregulation of protease-related genes probably eliminates the adverse effects of plant proteases in favor of A30 colonization at the late time point. Yet, the increase in subtilisin-like endopeptidases with a role in cuticle development [85] in the later stage proposes the evolution of a specialized trait to amplify these specific interactions. The PI-protease complex within the plant endophyte may be detected by host immune receptors and trigger defense responses, which reflects the arms race between microbes and their host plants [86].

## Host cell wall modifications modulate the *S. plymuthica* A30–tuber–*D. solani* interaction

The outcome of host-microbe interactions is determined by the dynamics of the plant cell wall. Alterations in cell wall dynamics can impact intracellular signal transduction and defense response at the site of pathogen invasion. In addition, both pathogenic and beneficial invaders activate plant cell wall remodeling not only by their own enzymes and proteins but also by manipulating the host's enzymes for successful colonization [87]. Our analysis uncovered the dual nature of cell wall modification in A30-treated tubers, where the timing of the interaction and the presence or absence of the pathogen can affect the outcome. The "defense side" is linked to the activation of genes related to crucial components of cell wall proteins, including extensins, arabinogalactan, and proline-rich proteins, contributing in cell wall signal transduction cascades and stress tolerance. In addition, A30-mediated cell wall defense involves the upregulation of genes associated with cell wall hydrolytic enzymes (including chitinases and glucanases), inhibitor proteins (including PEI and XTH inhibitor); and downregulation of genes related to degradation of host pectin and RG-degrading enzymes (polygalacturonase, pectate lyase, pectinesterase, and β-galactosidases) during the early phase of interaction. Simultaneously, the induction of callose synthases, cellulose synthases, and pectin synthases (galacturonosyltransferase) was accompanied by the upregulation of lignin and suberin-related genes. These changes correlate with the cell wall deposition at pathogen penetration sites, which serve as a resistance factor contributing to the strengthening of plant cell walls to restrict the pathogen spread and soft rot development. To adapt to the plant environment, endophytes can regulate plant cell wall-associated genes to establish a molecular invisibility cloak for successful colonization. In the late phase of interaction with A30, there was a transcriptional alteration of genes in cell wall remodeling, potentially increasing cell wall plasticity that probably

controls symplastic trafficking to facilitate intimate association with the host. These changes primarily belong to the activation of enzymes and proteins in cell wall loosening (including expansin-like protein), along with modification in cross-linking glycans, including XTH, which reassemble the structure of the cellulose-xyloglucan complex. Previous studies have highlighted the role of expansin-like proteins in plant-symbiotic interactions [88].

## Phytohormones JA and SA activate unique plant defenses in response to endophyte and pathogen

Our findings within the *D. solani*-potato pathosystem, consistent with our previous research [89] show early induction of SA-mediated defenses during the asymptomatic phase of infection, suggesting increased SAR as a defense response against *D. solani*. Subsequently, upon transition to the necrotrophic phase at later stages of infection, JA-mediated pathways become active, which is accompanied by a decrease in the initial level of endogenous SA, possibly due to the interplay between SA and JA. The reciprocal antagonism between SA and JA refers to the ambivalent effect they have on modulating the plant response to different stages of the pathogen's lifestyle [90]. In our study, the late activation of the JA-dependent defense mechanism serves as a response reaction to the production of oligogalacturonides (OGs) by *D. solani*. This occurs when a substantial bacterial population activates the QS system and releases high levels of degrading enzymes, resulting in tissue maceration during the necrotrophic phase of infection. Previous studies concluded the contribution of JA in defense-activating pectic fragments in the amplification of the DAMP response as a response to infection by soft rot bacteria [91,92].

The phytohormone analysis suggests that beneficial bacteria differentially affect hormonal status before and after infection by *D. solani*. We observed the early induction of the LOX pathway, leading to the production of the JA precursor cis-12-OPDA in A30-treated tubers both before and after pathogen challenge across early and late time points. However, activation of both the LOX and fatty acid β-oxidation pathways, responsible for converting OPDA to JA, were only triggered after pathogen infection during the early phase of interaction. In addition, the activation of the MYC branch of JA signaling by A30 at an early stage referred to the role of JA signaling in symbiosis and beneficial microbe-induced resistance [93], while the induction of the ERF branch of the JA pathway, which primarily responds to necrotrophs [49], was observed upon infection by *D. solani*. The rapid activation of JA-mediated defense by A30 in the early interaction stage, compared to the delayed JA-dependent response induced only with the pathogen in the later infection phase, could be a crucial factor in enhancing plant defense priming and recovery following the initial pathogen invasion. Surprisingly, while the elevated expression of LOX pathway-related genes in A30-treated tubers resulted in higher OPDA levels at later time points, this abundance of OPDA during the late interaction phase did not result in a rise in JA levels (Figs 5 and 6). As the activation of the JA-mediated pathway typically occurs during the symptomatic phase of *D. solani* infection in tubers, the absence of OPDA conversion to JA in the late response to the combination of *D. solani* and A30 may be attributed to the suppression of early infection events by the bacterial antagonist. The reduction in the pathogen population level in treated tubers by A30 at the late phase of interaction likely caused the inactivation of the QS system, preventing the conversion of OPDA into JA in the plant when there is no challenge or infection to defend against. It has been proposed that OPDA and other oxylipins in the plant play intrinsic roles in activating and fine-tuning defense responses mediated by beneficial microbes. For example, Wang and colleagues [94] detected OPDA in the xylem sap obtained from maize leaves with activated ISR by the beneficial fungus *Trichoderma virens*. OPDA acts as a long-distance signal, transmitting ISR defense

signals across the plant by regulating specific genes involved in protease inhibitors, biosynthesis of other hormones including ABA and ET, and promoting the production of glutathione and ROS to enhance callose deposition, which restricts pathogen spread [95]. However, OPDA can extend dormancy by modulating the expression of genes associated with ABA-sensing ABI5, GA-sensing *RGL2* DELLA, and the dormancy marker *MFT1* [96,97].

SA synthesis and SA-mediated defense in potato tubers were suppressed by strain A30 during both early and late responses. This suppression in SA levels led to an increase in the inactive form of SAG, notably observed at 168 hpi. The decline in SA supports the potential inhibitory crosstalk between JA and SA signaling, underscoring the role of JA in preventing excessive SA-mediated defense signaling during the early stages of interaction. This regulatory mechanism avoids the abortion of the symbiotic colonization event, a recognized phenomenon observed in plant-symbiotic microbe interactions [98]. Similar occurrences are observed in other endophytes, which reduce endogenous SA levels, likely due to trade-offs between SA and JA signaling [99,100]. Our findings highlight the importance of JA-mediated suppression of SA-associated defense pathways for the successful colonization of A30. This suppression induces mild stress, keeping the host in a quiescent state during symbiotic development. Similar JA-mediated immunosuppression for successful host colonization has been noted in other symbiotic microbes like *Piriformospora indica* [101] and root commensal bacteria [102].

### Involvement of ABA and ET in the regulation of tolerance and tuber dormancy

In our results, the alteration observed in ET-responsive genes may be a direct effect of strain A30. The early activation of genes involved in ET biosynthesis is consistent with previous findings suggesting the role of JA/ET signals in beneficial microbe-induced resistance. However, the upregulation of ACC deaminase in A30-treated tubers might lower host ET levels by scavenging the ET precursor ACC into ammonia and α-ketobutyrate [47]. These compounds can serve as carbon and nitrogen sources for endophytic microbes, promoting their growth and enhancing symbiosis by dampening the host's defense system. This phenomenon has been reported in numerous ACC deaminase-producing rhizobacteria [103–105], and their application has been shown to protect plants from different environmental stress conditions. From the data obtained at 168 hpi, the increase in gene expression related to ET biosynthesis and the lack of ACC deaminase expression implies an elevation in ET levels in A30-treated tubers, which subsequently return to normal levels after the stress diminishes. This rise in ET levels may affect the regulation of tuber dormancy by increasing ABA levels and inhibiting potato tuber sprouting [106,107]. The impact of ET on tubers varies depending on its concentration. In storage conditions, low concentrations of ET initially accelerate sprouting, while longer-term exposure to ET is believed to inhibit sprouting and extend dormancy [106].

The stress hormone ABA controls the proliferation and spreading of the pathogen through the negative impact on plant resistance [108], although there are exceptions, notably with some necrotrophic pathogens. For example, ABA in mimicking β-aminobutyric acid (BABA) triggers callose deposition at early defense responses against the necrotrophs, which is required for JA-dependent defense response [109,110]. Emerging evidence also indicates that ABA is crucial in establishing plant-microbe symbiotic relationships [111,112]. Moreover, ABA participates in priming the plant's immune response facilitated by beneficial microbes, with the outcome depending on the specific beneficial strain and the lifestyle of the pathogen. For example, ABA signaling components were found to be the key factors in both *P. fluorescens* and *B. subtilis*-mediated ISR against *Pst* DC3000 but not against *B. cinerea* [21]. On the other hand, ABA is actively involved in wound-induced suberization of potato tubers. An increase

in ABA-related gene expression was observed within 3 h of wounding and remained elevated through 96 h [113]. Numerous studies have documented the involvement of ABA when plants and their symbiotic partners encounter abiotic stresses [114,115]. The activation of ABA biosynthesis confirms its role in both the early and later responses of tubers to strain A30, both before and after the challenge with the pathogen. A significant increase in ABA levels in A30-treated tubers was observed in all treatments, with an almost three-fold accumulation at 168 hpi compared to the respective samples at 24 hpi. Collectively, we hypothesized that ABA might promote the colonization of potato tuber tissue by strain A30, similar to other symbiotic relationships, or prioritize tuber adaptation to existing abiotic stresses including wounding and cold storage conditions, by developing the formation of protective layers composed of suberin and callose. Stec et al. [112] proposed that ABA may serve as a determining factor in whether the host plant invests its resources in developing stress tolerance responses or establishing symbiotic interactions. ABA biosynthesis and signaling also play an important role in dormancy induction and maintenance in potatoes [116]. During the late response to A30, the significant increase in ABA levels, along with the noticeable induction of key components in ABA signaling hint at ABA's potential role in tuber dormancy. These changes were accompanied by alterations in the expression of genes associated with ABA-mediated dormancy or the negative regulation of seed germination, including *MFT1*, protein FRIGIDA, dormancy-associated MADS-box factors, and *HVA22*. As well as, the induction of drought and osmotic stress-responsive genes in an ABA-dependent manner, along with the upregulation of late embryogenesis abundant proteins *Lea5*-like and HSPs genes known for their involvement in acquiring drought tolerance during seed dormancy [117,118], may potentially elevate the level of dormancy in tubers treated with strain A30.

## Changes in auxin, GA, and CYT regulate the plant growth for successful colonization of A30

The alteration in IAA levels can be attributed to either beneficial or pathogenic microbes, either through modulation of the host auxin biosynthesis pathway or by direct production. The pathogen may employ auxin as a strategy to evade the plant defense system by suppressing SA-mediated defense responses or inducing cell wall loosening to facilitate penetration of the plant cell wall and access to nutrients [119]. Conversely, auxin signaling is closely linked to plant resistance and can be influenced by plant-colonizing microorganisms. Elevated cellular auxin levels can resemble a bacterial infection, tricking the plant into sensing an impending pathogen attack. Consequently, this may trigger an early priming cascade in response to the anticipated pathogen challenge, leading to enhanced ISR resistance [120,121]. The same principle could apply to *S. plymuthica* A30 as an IAA-producing endophytic bacterium. Auxin is a reciprocal signaling molecule in host-symbiotic interactions involved in multiple mechanisms, including nitrogen fixation, phosphate solubilization, and deaminase activity, known as phytostimulation [122]. In various plant-microbe interactions, plant symbionts have evolved to manipulate auxin transport for their purposes. For example, both efflux transporters PIN and AUX/LAX are required for the interaction of *Trichoderma virens* with *A. thaliana* [123], for nodulation in legume-rhizobium Interaction [124] and the regulation of root growth-promoting effect of *Serendipita indica* (synonym *Piriformospora indica*) [125]. In our experiment, induction of *PIN4*, *LAX1*, and *ABP1* suggest the contribution of these carrier proteins in auxin uptake during an endophytic relationship. On the other hand, prior research has shown that elevated levels of endogenous auxin in potato tubers prolong the dormancy, suppress sprouting, and lead to increased ET biosynthesis, while lower auxin levels promote sprout growth [126]. Another report revealed that IAA requires ABA in the maintenance of tuber dormancy.

*ABI5* and *ABI3* regulate the downstream genes of the auxin signaling (namely SUAR and AUX) and in turn, auxin stimulates *ABI3* expression through *ARF10* and *ARF16* to regulate the seed dormancy in synergy with ABA [127]. Taken together, increasing auxin and ABA levels accompanied by high level of ET at the late time point 168 hpi, may help in dormancy prolongation and inhibition of sprouting in A30-treated tubers.

Our data indicate that the upregulation of genes related to CK biosynthesis in the early time points in co-inoculated samples treated with both bacteria. Potentially, pathogens might manipulate host CK levels to modify plant physiology for their advantage including facilitating nutrient translocation. On the contrary, plants sense CK as an infection signal and promote plant defense against necrotrophic pathogens. This activation of CK indirectly triggers JA responses, leading to the deposition of callose [128,129]. In addition to regulating plant CK, some endophytic bacteria with PGPR activity can produce CKs which serve as bio-stimulant for plant growth or biocontrol against pathogens. For example, the produced CK by *P. fluorescens* efficiently primed the defense response against hemibiotrophic *P. syringes* in Arabidopsis [130,131]. GAs and CKs regulate dormancy cessation and initiation of tuber sprouting while both ABA and ET are required for dormancy maintenance [132]. In our data, growth hormones GA and CK were suppressed during the late response, suggesting inhibition of sprout growth. GA level decreased in A30-treated tubers due to the upregulation of negative GA signaling components DELLA, RGL2, and SCLs. Induction of DELLA and RGLs influences seed dormancy by stimulating *ABI5* activity to enhance the ABA-mediated repression of seed germination [133]. The decrease in CK biosynthesis was evidenced by the downregulation of zeatin biosynthesis and its derivatives leads to the suppression of tuber bud development.

## Impact of *S. plymuthica* A30 on primary and secondary metabolism of the host

The universal downregulation of photosynthesis is an adaptive response to perception of M/PAMP signal in both compatible and incompatible interactions in many pathosystem [134]. The photosynthesis and immune defense processes are interconnected, and the degree of photosynthetic changes could be an indicator of resistance level due to its role as the primary generator of ROS and defense components. Downregulation of most photosynthesis-related genes suggests that A30 shifts the metabolic prioritization of the tubers from photosynthesis to the production of other materials necessary for endophytic colonization or production of defense molecules. Induction of genes related to mitochondrial ETC provided clues to elucidate the role of endophyte A30 in mitochondrial cell activity (and hence, cellular energy supply) and deliberately ROS production as its main mechanism of action in the early hours after tuber treatment.

The plant's ability to mount an efficient defense deeply depends on the modulation of sugar efflux between the host and the pathogen [135]. In our study, many genes associated with carbohydrate metabolism and transport exhibited differential expression during the early and late responses to A30. The induction of genes involved in sugar distribution and transport in the early stages may supply nutrients and water, creating a suitable environment for both endophytic and pathogenic growth. This process is also supplying free sugars to fuel the energy required for the host defense mechanisms. In the early stages, A30 induced *SWEET*, invertase, and *PHO1*, promoting nutrient flow and facilitating endophytic colonization in tubers. Sugar transport SWEETs play a central role in plant-pathogen interaction [135]. Some studies have shown that beneficial microbe like pathogenic ones induce SWEET expression during plant colonization, for example, induction of Medicago *SWEET11* by rhizobia bacteria for the symbiotic nodule formation [136] and *SWEET1b* by AM fungus [137]. In the late response to A30,

the expression of *SWEET* and starch degradation genes was downregulated, coupled with an upregulation of invertase inhibitors in tubers. These findings suggest a potential suppression of sucrose mobilization and other carbohydrates, which, in turn, could inhibit tuber sprouting.

Our data revealed the induction of DEGs within lipid categories, including fatty acid elongation, unsaturated fatty acid biosynthesis, phospholipase-mediated fatty acid degradation, lipid oxidation, and alpha-linolenic acid metabolism in the early stage of interaction, which can support the biosynthesis of substrates including oxylipins, jasmonates and waxes. In addition, induction of marker genes expression involved in lipid beta-oxidation as well as galactolipids *MGDG* and *DGDG* were correlated with enhanced level of OPDA and JA in tubers co-inoculated with A30 and pathogen during the early stage of interaction. It has been demonstrated that an increased ratio of *MGDG*:*DGDG* induces JA overproduction [138]. In the late response to A30, downregulation of peroxisomal 3-ketoacyl-CoA thiolase in the final step of the β-oxidation pathway along with several fatty acid elongation enzymes (including *KAR*, *HACD*, and *ECR*) involved in wax production, may suggests a reduction in JA level and wax biosynthesis. This modulation in lipid metabolism, particularly at the intersection of wax production serving as a physical defense and JA-mediated defense in the A30-treated tuber, likely contributes to circumventing plant defenses and facilitating successful colonization.

Treatment with strain A30 affected the host's amino acid biosynthesis and transport, leading to changes in the biosynthesis phytohormones and the production of metabolites with antimicrobial properties. Among them, the induction of cysteine and methionine may participate in the production of sulfur-containing compounds with antioxidant or phytohormonal activity, and the induction of proline may promote tolerance against abiotic stresses and defense mechanisms against pathogens. These effects were also observed in other endophyte-plant interactions. For example, endophytic strains of Arthrobacter sp. and Bacillus spp. stimulate pepper plants to accumulate proline in response to osmotic stress [139], and high levels of proline during nodulation in legume-Rhizobium symbiosis are found specifically under stress conditions [140]. Our results suggest that A30 treatment triggered the expression of genes associated with polyamine biosynthesis, including agmatine, putrescine, spermidine, and spermine. Inoculation with *P. putida* GAP P45 modulated the expression of proline metabolic genes in *A. thaliana* under water stress conditions [141] and caused significant fluctuations in the production of putrescine, spermidine, and spermine, which was positively correlated with tolerance to water stress [142]. The role of polyamines has been demonstrated in several rhizobia-legume symbioses during the nitrogen fixation process [143–145]. Furthermore, the upregulation of NPF transporter suggests that A30 is involved in cellular transport and vacuolar sequestration of metabolites, attributed to this transporter family. Induction of nitrate transporter in tomato reduces ion toxicity in PGPR-mediated salt tolerance and transports nitrate into the fruit for subsequent defense against potential invaders [146,147].

The interaction between host and endophyte is tightly controlled by secondary metabolism. The metabolites and physiological states of the host plant can be affected by endophytes directly or indirectly in a variety of ways. Endophyte influences host metabolites for its successful entry and spreading inside the plant, and also boosts the defensive pathways against subsequent invaders [15]. During the early stage of interaction, strain A30 activated defense-related metabolic reprogramming in tubers by changes in expression levels of genes involved in the production of phenylpropanoids, benzoates, flavonoids, sesquiterpenoids, and glycoalkaloids. These include the production of polyphenol compounds including monolignols for cell wall lignification, phenolic monomers in suberization affecting physical resistance, flavonoids, and hydroxycinnamate derivatives with antioxidant or phytoalexin properties as a chemical defense. Other than, A30 affect the production of sesquiterpenoid phytoalexins solavetivone and rishitin, and most likely also other bioactive hydroxamides, phenolics and

stilbenoid compounds that play an important role in the biocontrol of plant diseases, providing evidence of chemical defense against *D. solani* infection mainly in the early stage of interaction. It has been previously indicated that hydroxamic acid DIMBOA present in grass species has a bactericidal and bacteriostatic effects on soft rot bacteria [148]. Colonization of maize roots by *P. putida* triggers defense priming by emission of terpenoid volatiles and by priming serine proteinase inhibitor, which depends on benzoxazinoids from root exudation [149]. Furthermore, a decrease in glycoalkaloid-related genes at 168 hpi implies the potential efficacy of the biocontrol agent A30 in lowering solanine levels. This phenomenon has been observed with other biological control agents, such as *Bacillus megaterium*, which exhibits antimicrobial properties against *Pectobacterium atrosepticum* and suppresses solanine in potato tubers [150]. Similarly, *Bacillus subtilis* has been shown to regulate solanine levels in both healthy and *Fusarium*-infected potato tubers during storage [151]. Despite downregulation of potato tuber defense during late interaction, the increased in biosynthesis of secondary metabolites may serve as a defense strategy against the pathogen and as a possible endophyte competitive exclusion tactic. Additionally, the decrease in plant immunity reallocates significant energy resources towards primary or secondary plant metabolism.

## Conclusion

This study aimed to uncover molecular dynamics underlying the responses triggered by *S. plymuthica* A30 in potato tubers, both infected and non-infected with *D. solani*, during the early and late phases of interaction (Fig 8). In natural settings, *S. plymuthica* A30 induces early-

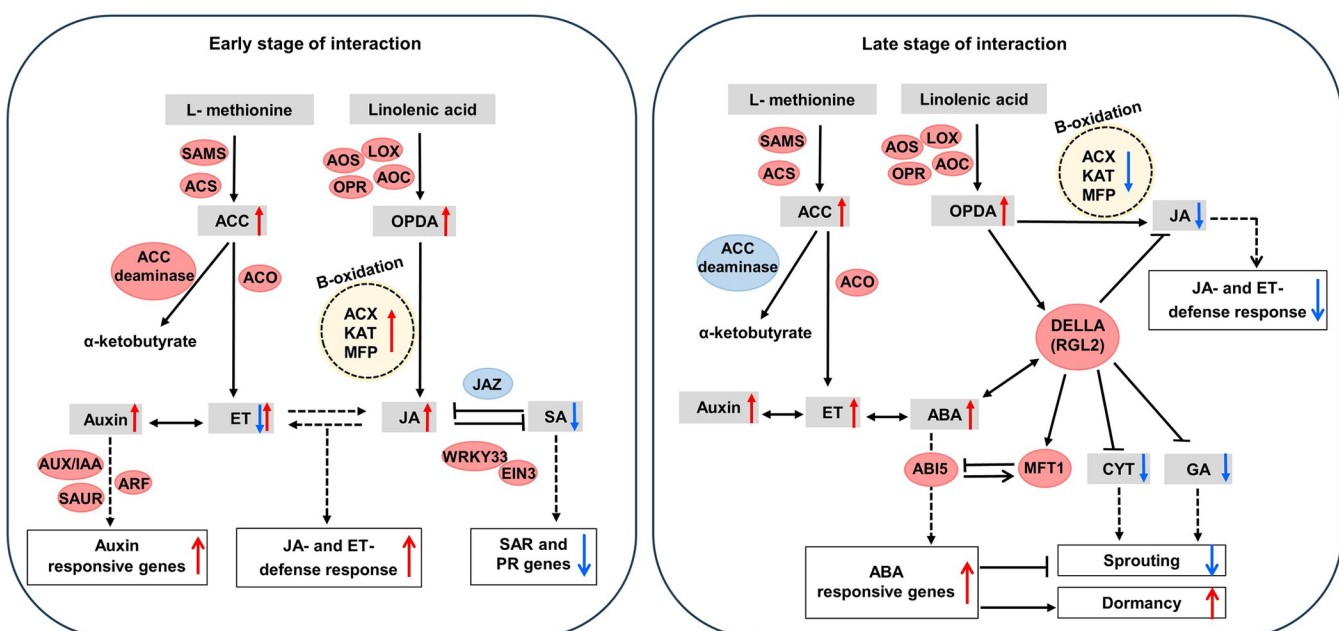

**Fig 8. . Proposed hormonal crosstalk and involved key genes regulating the potato tuber response to combined inoculation with *D. solani* and *S. plymuthica* A30 during early and phases of interaction.** The figure depicts genes (circles) and compounds (rectangles) involved in the hormonal crosstalk. Upregulated genes are shown in red circles, while downregulated genes are shown in blue circles. Red arrows within rectangles indicate positive regulation by a compound, while blue arrows indicate negative regulation. Broken lines represent indirect regulatory processes. Abbreviation used: LOX, lipoxygenase; AOS, allene oxide synthase; AOC, allene oxide cyclase; OPR, 12-oxophytodienoate reductase; OPDA, 12-oxo-phytodienoic acid; ACX, acyl CoA oxidase; KAT, 3-ketoacyl-CoA-thiolase; MFP, multifunctional protein; JAZ, jasmonate ZIM domain; EIN3, ethylene insensitive 3; SAR, systemic acquired resistance; PR, pathogenesis-related genes; SAMS, S-adenosyl-methionine (SAM) synthase; ACC, 1-aminocyclopropane-1-carboxylic acid; ACS, ACC synthase; ACO, ACC-oxidase; ARF, auxin response factor; SAUR, small auxin up-regulated RNA; ABI5, abscisic acid insensitive 5; RGL2, RGA (repressor of gibberellin response 1)-like 2; MFT1, mother of FT and TFL1.

phase plant responses akin to those of known beneficial endophytic microbes, while also partially suppressing the expression of defense-related genes in later phase, facilitating successful and enduring colonization of the host. In the presence of *D. solani*, biocontrol strain A30 activated a substantial recognition system around the infection site, contributing to the rapid achievement of the signaling threshold required to stimulate the plant immune system. Diverse biological processes are integrated to prioritize defense mechanisms against infection, orchestrating plant responses through both physical and chemical barriers. Furthermore, A30-induced resistance relied on the early activation of JA signaling and its production specifically in tubers inoculated with both bacterial strains. The bacterium A30 which originally found within potato tubers, forms a symbiotic relationship that confers multiple advantages to the host plant, including protection against soft rot disease and extension of dormancy periods, through induction of dormancy-related hormones (ABA and ET) and suppression of growth-related hormones (GA and CK), thereby minimizing losses and enhancing storage quality. Our study has identified genes that could serve as valuable resistance inducers in genetic engineering research to provide a promising strategy for controlling soft rot in potato tubers. This study sheds light on how beneficial bacteria regulate potato defenses response against soft rot pathogens and identifies biomarkers that influence induced resistance, crucial for improving biocontrol efficacy.

## Supporting information

**S1 Table. All Identified transcripts and list of differentially expressed genes in time course RNA-Seq.** Transcripts at 1-, 24- and 168-hours post-inoculation in time course RNA-Seq data of potato tubers inoculated with *Serratia plymuthica* with or without *Dickeya solani* identified with NOISeq with p-value $< 0.05$ and $-2 \geq$ log 2-fold change $\geq 2$.
(XLSX)

**S2 Table. The list of categorized differentially expressed genes of time course RNA-Seq.** The list of categorized genes of time-course RNA-Seq data for 1-, 24- and 168-hours post-inoculation that identified with NOIseq with p-value $< 0.05$ and $-2 \geq$ log 2-fold change $\geq 2$.
(XLSX)

**S3 Table. All identified genes and list of differentially expressed genes in second RNA-Seq.** All identified genes and the list of differentially expressed genes of second RNA-Seq data for -A30+D and +A30+D that identified with NOISeq with p-value $< 0.05$ and $-2 \geq$ log 2-fold change $\geq 2$.
(XLSX)

**S4 Table. The list of primers and qPCR data of both RNA-Seq analysis.** qRT–PCR validation of differentially expressed genes (DEGs) obtained from both RNA-Seq of potato tubers inoculated with *D. solani* and/or *Serratia plymuthica* A30. Data were obtained from three independent cDNA sets from three independent experiments, normalized to eukaryotic elongation factor 5A3 and expressed as the means of log2 (ΔΔCt) ± SEM (standard error of the mean). The graphs demonstrate the means of the log2 fold changes from three independent experiments. Error bars show the standard error of the mean, and statistical tests were performed with t-tests. Primers were used in this study include primers for 34 DEGs from the time course RNA-Seq and for 15 DEGs from the second RNA-Seq, along with the target genes IDs, abbreviations and primer sequences.
(XLSX)

## Acknowledgments

We would like to thank Beijing Genome Institute for their helping to analysis of transcriptomes data. Open access funded by Helsinki University Library.

## Author Contributions

**Conceptualization:** Iman Hadizadeh, Bahram Peivastegan.

**Data curation:** Kåre Lehmann Nielsen, Petri Auvinen.

**Formal analysis:** Iman Hadizadeh, Bahram Peivastegan, Nina Sipari.

**Investigation:** Iman Hadizadeh, Bahram Peivastegan.

**Project administration:** Minna Pirhonen.

**Resources:** Kåre Lehmann Nielsen, Petri Auvinen, Minna Pirhonen.

**Supervision:** Minna Pirhonen.

**Validation:** Iman Hadizadeh, Bahram Peivastegan.

**Visualization:** Iman Hadizadeh, Bahram Peivastegan.

**Writing – original draft:** Iman Hadizadeh, Bahram Peivastegan.

**Writing – review & editing:** Iman Hadizadeh, Bahram Peivastegan.

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
