## [Decision Letter · Decision Letter 0]

27 May 2024

PONE-D-24-10853Transcriptome analysis unravels the biocontrol mechanism of Serratia plymuthica A30 against potato soft rot caused by Dickeya solaniPLOS ONE

Dear Dr. Hadizadeh,

Thank you for submitting your manuscript to PLOS ONE. After careful consideration, we feel that it has merit but does not fully meet PLOS ONE’s publication criteria as it currently stands. Therefore, we invite you to submit a revised version of the manuscript that addresses the points raised during the review process.

We look forward to receiving your revised manuscript.

Kind regards,

Kandasamy Ulaganathan

Academic Editor

PLOS ONE

Reviewers' comments:

Reviewer's Responses to Questions

**Comments to the Author**

1. Is the manuscript technically sound, and do the data support the conclusions?

Reviewer #1: Yes

Reviewer #2: Yes

2. Has the statistical analysis been performed appropriately and rigorously? 

Reviewer #1: Yes

Reviewer #2: Yes

3. Have the authors made all data underlying the findings in their manuscript fully available?

Reviewer #1: Yes

Reviewer #2: Yes

4. Is the manuscript presented in an intelligible fashion and written in standard English?

Reviewer #1: Yes

Reviewer #2: Yes

5. Review Comments to the Author

Reviewer #1: Hadizadeh et al. manuscript entitled “Transcriptome analysis unravels the biocontrol mechanism of Serratia plymuthica A30 against potato soft rot caused by Dickeya solani” looks very interesting and technically sound. This type of study should be recommended for publication. However, there are some points requiring consideration.

1. The Abstract should reflect the problem, the method, and the conclusions, please further improve

2. In the last paragraph of the Introduction, please state your motivation, goals and objectives, the novelty of this research, and potential contribution to the literature.

3. Make changes in Keywords; because most of the words are included in the paper title.

4. In page 6, on lines 203-206, mention the software and the corresponding parameters used for preprocessing of the sequenced data.

5. In page 6, on line 218, mention the version of NOISeq used for the analysis.

6. In page 7, on line 250, write “ p ” instead of “ P ”. p-value should always be written as “p” or “p-value”. Correct it throughout the manuscript.

7. All the figures in the manuscript are of very poor quality making it unable to read the labels. They should be replaced with the high-resolution images.

8. Authors effort to illustrate the role of plant hormones SA, JA, ethylene, ABA, Auxin, cytokinin and GA is impressive. Figure 8 illustration on proposed defense response mechanism of potato tuber towards S. plymuthica A30 and infection by D. solani clearly portrayed the complex responses and interplay of several plant hormones. It would be better to include the same in the manuscript under separate section with a brief discussion.

9. Conclusion can be improved further.

10. Check the manuscript for all the typo errors and correct accordingly.

Reviewer #2: The manuscript "Transcriptome analysis unravels the biocontrol mechanism of Serratia plymuthica A30 against potato soft rot caused by Dickeya solani" (PONE-D-24-10853) submitted by Hadizadeh et al describes the transcriptional response of potato tubers to inoculation with the biocontrol agent S. plymuthica A30, pathogen Dickeya solani or a combination of both. The objective was "to uncover molecular dynamics underlying the responses triggered by S. plymuthica A30 in potato tubers, both infected and uninfected with D. solani", and in this regard the authors are generally successful. The data presented is comprehensive and presented in a systematic and methodical manner. The main conclusions, that A30 effectively primes the plant immune system to better respond to pathogen infection seems supported by the data, though is not a particularly novel finding. There are a number of issues that the authors should address to enhance the impact of their manuscript:

1. In the materials & methods section, the timing of tissue collection post inoculation is not clear. For example, on lines 159-162 the authors write "First, a group of tubers was inoculated with 25 μl suspension of strain S. plymuthica A30 and another group with 25 μl of sterile water as non-inoculated control. After two hours, non-inoculated controls and tubers treated with strain A30 were inoculated with 25 μl suspension of D. solani or an extra 25 μl of water, depending on the treatments". Then on lines 186-187, the authors write "In the first RNA-Seq experiment, the time course experiment was performed using the Illumina HiSeq 2000 platform with 18 samples consisting of three time points 1, 24, and 168... but it is not clear if this is after the second stage of inoculation. If so, this is already three hours after wounding, which can potentially complicate the analysis as wounding itself triggers a massive change in gene expression in tubers (see Woolfson et al., 2023, Phytochemistry https://doi.org/10.1016/j.phytochem.2022.113529). The authors do suggest that covering the wounds with Vaseline was done to avoid the healing process (lines 163/64), but what is the evidence for this?

2. The discussion is largely a re-hash of the results. It isn't until the "Conclusion" section that the authors attempt to compile a model of what may be happening in the A30-D. solani co-inoculation. And while Figure 8 does provide a good summary, it is a very complex composite. The caption does not provide enough detail to guide readers through the temporal aspect of the interaction. For example, the first part of Figure 8 depicts the early stages of the interaction, but includes things such as lignification and suberization modifications to the cell wall. While biosynthesis of the monomers for these polymers may be initiated at early stages, their deposition takes several days and would not be captured within the first 24 hours. The authors may consider breaking this figure into smaller segments based on the transcriptional analysis at each time point.

3. Related to point 1 above, wounding of tubers triggers phenylpropanoid metabolism and suberization. But the authors suggest that their treatment of of the tissue supressess the wound response. Do they suggest then that there is a separate process for the initiation of suberization via microbial interation that circumvents the wound initiation of these processes?

6. PLOS authors have the option to publish the peer review history of their article (what does this mean?). If published, this will include your full peer review and any attached files.

Reviewer #1: **Yes: **Dr. Sravanthi Burragoni

Reviewer #2: **Yes: **Mark A. Bernards

---

## [Author Response · Author response to Decision Letter 0]

9 Jul 2024

Dear Editor,

We would like to thank you and the reviewers for your thoughtful and constructive comments on our manuscript. We have carefully considered each point raised and have made the necessary revisions to address these concerns. Below, we provide a detailed response to each comment.

Dear Dr. Sravanthi Burragoni (Reviewer 1),

We thank you for your valuable comments and suggestions, which have significantly improved our manuscript. Below are our responses to each point raised.

1. The Abstract should reflect the problem, the method, and the conclusions, please further improve.

Response: We have revised the Abstract to effectively convey the problem statement, the methodology employed, and the conclusions drawn from our study. The updated Abstract now provides a concise summary of these elements. 

2. In the last paragraph of the Introduction, please state your motivation, goals and objectives, the novelty of this research, and potential contribution to the literature. 

Response: We have revised the last paragraph of the Introduction to explicitly articulate our motivations, aims, and objectives. Additionally, we have emphasized the unique aspects of our research and its potential impact on the current body of knowledge in this field.

3. Make changes in Keywords; because most of the words are included in the paper title. 

Response: We have updated the Keywords to include terms that are not present in the title, ensuring they complement the title and enhance the discoverability of our manuscript.

4. In page 6, on lines 203-206, mention the software and the corresponding parameters used for preprocessing of the sequenced data. 

Response: We have added the specific software used for preprocessing the sequenced data along with the corresponding parameters in lines 203-206 on page 6.

5. In page 6, on line 218, mention the version of NOISeq used for the analysis. 

Response: The version of NOISeq used for the analysis has been included on line 218 of page 6. We used NOISeq version 2.28.0 for our differential expression analysis.

6. In page 7, on line 250, write “p” instead of “P”. p-value should always be written as “p” or “p-value”. Correct it throughout the manuscript. 

Response: We have corrected “P” and “P-value” to “p” or “p-value” throughout the manuscript, including the specific correction on line 250 of page 7.

7. All the figures in the manuscript are of very poor quality making it unable to read the labels. They should be replaced with high-resolution images. 

Response: All figures have been replaced with high-resolution images to ensure that the labels and details are clearly readable.

8. Authors effort to illustrate the role of plant hormones SA, JA, ethylene, ABA, Auxin, cytokinin and GA is impressive. Figure 8 illustration on proposed defense response mechanism of potato tuber towards S. plymuthica A30 and infection by D. solani clearly portrayed the complex responses and interplay of several plant hormones. It would be better to include the same in the manuscript under separate section with a brief discussion. 

Response: Thank you for your positive feedback on the illustration of plant hormone roles and the proposed defense response mechanism in Figure 8. Based on your suggestion, we have revised the figure to focus solely on the hormonal crosstalk at early and late time points. This simplified figure has been included in the conclusion section to provide a clearer depiction of the hormonal interactions. We believe this change enhances the clarity and focus of our manuscript. Thank you again for your valuable input.

9. Conclusion can be improved further. 

Response: The Conclusion section has been revised to more effectively summarize the findings and implications of our research.

10. Check the manuscript for all the typo errors and correct accordingly. 

Response: We have thoroughly checked the manuscript for typographical errors and have made the necessary corrections.

We trust that these revisions align with the reviewers' expectations and further improve our manuscript. Thank you for considering our revised submission.

Sincerely,

Dear Professor Mark A. Bernards (Reviewer 2),

We greatly appreciate your constructive feedback and recommendations, which have profoundly enhanced the quality of our manuscript. Below, we provide detailed responses to each of the points you raised.

Comment 1: In the materials & methods section, the timing of tissue collection post-inoculation is not clear. For example, on lines 159-162 the authors write "First, a group of tubers was inoculated with 25 μl suspension of strain S. plymuthica A30 and another group with 25 μl of sterile water as non-inoculated control. After two hours, non-inoculated controls and tubers treated with strain A30 were inoculated with 25 μl suspension of D. solani or an extra 25 μl of water, depending on the treatments". Then on lines 186-187, the authors write "In the first RNA-Seq experiment, the time course experiment was performed using the Illumina HiSeq 2000 platform with 18 samples consisting of three time points 1, 24, and 168... but it is not clear if this is after the second stage of inoculation. If so, this is already three hours after wounding, which can potentially complicate the analysis as wounding itself triggers a massive change in gene expression in tubers (see Woolfson et al., 2023, Phytochemistry https://doi.org/10.1016/j.phytochem.2022.113529). The authors do suggest that covering the wounds with Vaseline was done to avoid the healing process (lines 163/64), but what is the evidence for this?

Response: 

We apologize for the lack of transparency regarding the timing of tissue sampling post-inoculation. To clarify, the first RNA-Seq experiment involved sampling at 1, 24, and 168 hours post the second inoculation. We have revised the manuscript to clearly state this timing.

Regarding the use of Vaseline, its application was intended to maintain high humidity at the inoculation site and slow down the healing process induced by wounding. Preserving high humidity at the inoculation site is crucial for bacterial growth and population proliferation (Murant & Wood, 1959; Chen et al., 2020; Pérombelon, 2002). This is a well-established method employed by research labs working with soft rot bacteria in potato tubers.

Furthermore, this inoculation method and 2-hour pause between the inoculations of the two bacteria were applied consistently across all samples. This consistency allows for a valid comparison between the control and treatment group at each time point, as any observed differences can be attributed to the presence or absence of bacteria rather than variations in the wounding timing. Thus, the consistent wounding method and 2-hour pause between inoculations ensured uniformity across all samples. This approach minimized variability and potential differential effects, thereby enhancing the reliability and validity of the experimental results.

Comment 2: The discussion is largely a re-hash of the results. It isn't until the "Conclusion" section that the authors attempt to compile a model of what may be happening in the A30-D. solani co-inoculation. And while Figure 8 does provide a good summary, it is a very complex composite. The caption does not provide enough detail to guide readers through the temporal aspect of the interaction. For example, the first part of Figure 8 depicts the early stages of the interaction, but includes things such as lignification and suberization modifications to the cell wall. While biosynthesis of the monomers for these polymers may be initiated at early stages, their deposition takes several days and would not be captured within the first 24 hours. The authors may consider breaking this figure into smaller segments based on the transcriptional analysis at each time point.

Response: We appreciate the reviewer's insightful feedback on Figure 8 and the discussion section. Regarding the reviewer’s specific concern about lignification and suberization in the early stages of the interaction, we acknowledge that the deposition of these compounds takes several days. It is important to note that our discussion are based solely on transcriptome data, indicating the induction of genes involved in biosynthesis of suberin and lignin monomers. The reviewer is correct; our current manuscript focuses on the transcriptional response and presents evidence for the upregulation of genes associated with suberin and lignin biosynthesis. However, we have not yet obtained direct evidence for the accumulation of these cell wall components.

Furthermore, we agree that Figure 8 was complex and could benefit from simplification. In response, we have refined Figure 8 to exclusively highlight the hormonal crosstalk occurring at early and late stages, thereby providing a clearer and more focused depiction of hormone interactions during these distinct phases. The revised figure now clearly delineate the hormonal crosstalk during different interaction phases, enhancing the clarity of our model and addressing the reviewer's concerns.

Comment 3: Related to point 1 above, wounding of tubers triggers phenylpropanoid metabolism and suberization. But the authors suggest that their treatment of the tissue suppresses the wound response. Do they suggest then that there is a separate process for the initiation of suberization via microbial interaction that circumvents the wound initiation of these processes?

Response: 

Thank you for your insightful comment regarding the induction of phenylpropanoid metabolism and suberization in potato tubers. Our results indicate that the induction of the genes -related to biosynthesis of suberin monomers were specifically observed in potato tubers treated with both the pathogen (D. solani) and the antagonist (S. plymuthica A30) at early response, which were subsequently suppressed in the late response. The combined effect of the antagonist and pathogen in our experimental setup likely accelerates the expression of genes involved in biosynthesis of suberin and lignin monomers, preparing the cell wall defense. In our experiments, we did not observe significant induction of these genes when the tubers were treated only with bacterial antagonist. This suggested the presence of both bacteria (probably due to direct antagonist effect) accelerate the induction of genes related to several biological process including suberin and lignin biosynthesis. The induction of these genes does not necessarily leads to the immediate synthesis of suberin. It is reasonable that while the biosynthesis-related genes are upregulated early, the actual accumulation of suberin polymers takes longer time, as indicated in the literature (Woolfson et al., 2023; Woolfson et al., 2022). 

There is an evidence indicating that certain microbes can alter the production of plant secondary metabolites (such as suberin and lignin) in plant (Pang et al., 2021; Al-Khayri et al., 2023). For example, research has shown upregulation of suberin biosynthesis related genes such as CYP86A1, CYP86B1, and MYB107, along with suberin accumulation in the resistant rice when infected by nematode (Singh et al., 2021).

We hope these revisions address the reviewers' concerns satisfactorily. Thank you for your insightful comments and for the opportunity to improve our manuscript.

The references were used:

Murant, A. F. & Wood, R. K. S., 1959. Factors affecting the pathogenicity of bacteria to potato tubers. Ann. appl. Biol. 45 (4):650–663.

Chen, D., Nahar, K., Bizimungu, B. et al. A Simple and Efficient Inoculation Method for Fusarium Dry Rot Evaluations in Potatoes. Am. J. Potato Res. 97, 265–271 (2020). https://doi.org/10.1007/s12230-020-09774-4

Pérombelon, M.C.M. (2002), Potato diseases caused by soft rot erwinias: an overview of pathogenesis. Plant Pathology, 51: 1-12. https://doi.org/10.1046/j.0032-0862.2001.Shorttitle.doc.x

Woolfson KN, Esfandiari M, Bernards MA. Suberin Biosynthesis, Assembly, and Regulation. Plants. 2022; 11(4):555. https://doi.org/10.3390/plants11040555

Woolfson, K. N., Zhurov, V., Wu, T., Kaberi, K. M., Wu, S., & Bernards, M. A. (2023). Transcriptomic analysis of wound-healing in Solanum tuberosum (potato) tubers: Evidence for a stepwise induction of suberin-associated genes. Phytochemistry, 206, 113529. https://doi.org/10.1016/j.phytochem.2022.113529

Pang, Z., Chen, J., Wang, T., Gao, C., Li, Z., Guo, L., Xu, J., & Cheng, Y. (2021). Linking Plant Secondary Metabolites and Plant Microbiomes: A Review. Frontiers in plant science, 12, 621276. https://doi.org/10.3389/fpls.2021.621276

Al-Khayri JM, Rashmi R, Toppo V, Chole PB, Banadka A, Sudheer WN, Nagella P, Shehata WF, Al-Mssallem MQ, Alessa FM, et al. Plant Secondary Metabolites: The Weapons for Biotic Stress Management. Metabolites. 2023; 13(6):716. https://doi.org/10.3390/metabo13060716

Singh, D.P., Dutta, T.K., Shivakumara, T.N., Dash, M., Bollinedi, H., & Rao, U. (2021). Suberin Biopolymer in Rice Root Exodermis Reinforces Preformed Barrier Against Meloidogyne graminicola Infection. Rice Science, 28, 301-312.

---

## [Decision Letter · Decision Letter 1]

30 Jul 2024

Transcriptome analysis unravels the biocontrol mechanism of Serratia plymuthica A30 against potato soft rot caused by Dickeya solani

PONE-D-24-10853R1

Dear Dr. Hadizadeh,

We’re pleased to inform you that your manuscript has been judged scientifically suitable for publication and will be formally accepted for publication once it meets all outstanding technical requirements.

Kind regards,

Kandasamy Ulaganathan

Academic Editor

PLOS ONE

Additional Editor Comments (optional):

Reviewers' comments:

Reviewer's Responses to Questions

**Comments to the Author**

1. If the authors have adequately addressed your comments raised in a previous round of review and you feel that this manuscript is now acceptable for publication, you may indicate that here to bypass the “Comments to the Author” section, enter your conflict of interest statement in the “Confidential to Editor” section, and submit your "Accept" recommendation.

Reviewer #2: All comments have been addressed

2. Is the manuscript technically sound, and do the data support the conclusions?

Reviewer #2: Yes

3. Has the statistical analysis been performed appropriately and rigorously? 

Reviewer #2: N/A

4. Have the authors made all data underlying the findings in their manuscript fully available?

Reviewer #2: Yes

5. Is the manuscript presented in an intelligible fashion and written in standard English?

Reviewer #2: Yes

6. Review Comments to the Author

Reviewer #2: While the authors have revised their conclusions and summary figure 8, the discussion remains virtually unchanged and largely a rehash of the results.

7. PLOS authors have the option to publish the peer review history of their article (what does this mean?). If published, this will include your full peer review and any attached files.

Reviewer #2: **Yes: **Mark A. Bernards

---

## [Editor Report · Acceptance letter]

29 Aug 2024

PONE-D-24-10853R1 

PLOS ONE

Dear Dr. Hadizadeh, 

I'm pleased to inform you that your manuscript has been deemed suitable for publication in PLOS ONE. Congratulations! Your manuscript is now being handed over to our production team.

Kind regards, 

on behalf of

Dr. Kandasamy Ulaganathan 

Academic Editor

PLOS ONE